# Coordination of humoral immune factors dictates compatibility between *Schistosoma mansoni* and *Biomphalaria glabrata*

Hongyu Li[1,2], Jacob R Hambrook[2], Emmanuel A Pila[2], Abdullah A Gharamah[2], Jing Fang[1,2], Xinzhong Wu[1,3]*, Patrick Hanington[2]*

[1]Ocean College, Beibu Gulf University, Qinzhou, China; [2]School of Public Health, University of Alberta, Edmonton, Canada; [3]College of Animal Sciences, Zhejiang University, Hangzhou, China

**Abstract** Immune factors in snails of the genus *Biomphalaria* are critical for combating *Schistosoma mansoni*, the predominant cause of human intestinal schistosomiasis. Independently, many of these factors play an important role in, but do not fully define, the compatibility between the model snail *B. glabrata*, and *S. mansoni*. Here, we demonstrate association between four previously characterized humoral immune molecules; *Bg*FREP3, *Bg*TEP1, *Bg*FREP2 and Biomphalysin. We also identify unique immune determinants in the plasma of *S. mansoni*-resistant *B. glabrata* that associate with the incompatible phenotype. These factors coordinate to initiate haemocyte-mediated destruction of *S. mansoni* sporocysts via production of reactive oxygen species. The inclusion of *Bg*FREP2 in a *Bg*FREP3-initiated complex that also includes *Bg*TEP1 almost completely explains resistance to *S. mansoni* in this model. Our study unifies many independent lines of investigation to provide a more comprehensive understanding of the snail immune system in the context of infection by this important human parasite.

**\*For correspondence:**
wuxzqinzhou@163.com (XW);
pch1@ualberta.ca (PH)

**Competing interests:** The authors declare that no competing interests exist.

## Introduction

Schistosomiasis, a disease caused by parasitic trematodes of the genus *Schistosoma*, is second only to malaria as the most socioeconomically devastating parasitic disease, with an estimated 252 million people infected worldwide in 2015 (*Disease et al., 2016*; *Mortality and GBD 2015 Mortality and Causes of Death Collaborators, 2016*). Larval digenean trematodes share a common feature in the use of snails (gastropod mollusks) as intermediate hosts for transmission to a vertebrate host (*Esch et al., 2002*). The snail *Biomphalaria glabrata* transmits several species of trematode, including *Schistosoma mansoni*, the predominant causal species of intestinal schistosomiasis (*Pila et al., 2016a*). The molecular interactions between *B. glabrata* and *S. mansoni* have been studied extensively towards better understanding this essential life cycle stage of an important human parasite (*Adema, 2015*).

Various genetically determined resistance phenotypes exist with respect to *S. mansoni* infection in *B. glabrata* (*Richards, 1975a*; *Richards, 1975b*). Some *B. glabrata* snails (such as BS-90 strain) naturally resist *S. mansoni* infection. Susceptible *B. glabrata* (such as M-line strain) can develop transient levels of acquired resistance following previous exposure to incompatible or radiation-attenuated parasites with modest cross protection to related digenean species (*Lie and Heyneman, 1975*; *Lie et al., 1975*; *Sullivan et al., 1982*). Previous research has focused on identifying the mechanisms responsible for determining resistance profiles. Using in vivo and in vitro models of *B. glabrata* snails, it has been demonstrated that the killing of *S. mansoni* larvae is associated with a haemocyte-

mediated cytotoxic mechanism, and passive transfer of natural resistance to *S. mansoni* has been successfully accomplished when haemocytes from a susceptible *B. glabrata* strain are incubated in cell-free haemolymph (plasma) from a resistant strain (*Bayne et al., 1980a*; *Bayne et al., 1980b*; *Granath and Yoshino, 1984*; *Loker and Bayne, 1982*). Thus, haemocytes from resistant or susceptible *B. glabrata* strains do not appear to differ a priori in their cytotoxic capabilities, but their response requires activation by some humoral factor(s) (*Bayne et al., 1980b*; *Granath and Yoshino, 1984*; *Vasquez and Sullivan, 2001*), for proper recognition of *S. mansoni* and enhancing haemocyte cytotoxicity (*Hahn et al., 2001*).

Researchers have long sought immune determinants present in resistant *B. glabrata* plasma that specifically activate haemocytes to encapsulate and destroy *S. mansoni* sporocysts. Identified in *B. glabrata*, a large family of soluble lectins termed fibrinogen-related proteins (*Bg*FREPs) attracted widespread interest because of their unique structure (*Adema et al., 1997*; *Gordy et al., 2015*), capacity for somatic diversification (*Zhang et al., 2004*), and immune function (*Hanington et al., 2010*; *Hanington and Zhang, 2011*). *Bg*FREPs are composed of a C-terminal fibrinogen (FBG) domain and either one or two N-terminal immunoglobulin superfamily (IgSF) domains linked to the FBG domain by an interceding region (ICR) (*Léonard et al., 2001*; *Zhang et al., 2001*). Numerous studies have implicated *Bg*FREP3 as playing a central role in *B. glabrata* resistance to digenetic trematodes (*Hanington et al., 2010*; *Hanington et al., 2012*; *Hanington and Zhang, 2011*; *Lockyer et al., 2012*; *Lockyer et al., 2008*; *Pila et al., 2017b*). *Bg*FREP3 contains two tandem IgSF domains whereas *Bg*FREP2 has one (*Léonard et al., 2001*). The expression of *Bg*FREP2 in BS-90 snails is dramatically up-regulated (57-fold) after exposure to *S. mansoni*, whereas M-line snails do not feature such a drastic upregulation, thereby suggesting a role for *Bg*FREP2 in resistance (*Hertel et al., 2005*). Interestingly, the time course for heightened *Bg*FREP2 expression overlaps the interval within which sporocysts are encapsulated and killed, and only occurs in BS-90 haemocytes (*Dinguirard et al., 2018*; *Hertel et al., 2005*). Furthermore it was reported that *Bg*FREP2 is reactive with *S. mansoni* polymorphic mucins (*Sm*PoMucs) found on the tegumental surface of larval *S. mansoni* (*Moné et al., 2010*).

In addition to *Bg*FREPs, other soluble immune effector factors which are involved in both the direct killing of sporocysts and preparation of haemocytes to mount a cell-mediated response have been characterized (*Pila et al., 2017a*). These factors include *B. glabrata* thioester-containing proteins (*Bg*TEPs) (*Bender et al., 1992*; *Moné et al., 2010*; *Portet et al., 2018*) and Biomphalysin (*Galinier et al., 2013*). TEPs are key components of invertebrate and vertebrate immune responses, aiding in melanization, opsonization, and killing of invading pathogens (*Baxter et al., 2007*; *Blandin et al., 2008*; *Levashina et al., 2001*; *Pila et al., 2017a*; *Portet et al., 2018*; *Povelones et al., 2013*; *Yassine and Osta, 2010*). *Bg*TEP was first characterized as an alpha macroglobulin proteinase inhibitor, and has recently re-emerged as a molecule of interest due to its association with *Bg*FREP2 and *Sm*PoMucs, and the capacity to bind to various invading pathogens (*Bender et al., 1992*; *Moné et al., 2010*; *Portet et al., 2018*). Biomphalysin is a cytolytic protein in *B. glabrata* belonging to the β pore-forming toxin (β-PFT) superfamily (*Galinier et al., 2013*). Biomphalysin binds to the surface of *S. mansoni* sporocysts in the absence of plasma, while its cytolytic activity is drastically increased when plasma is present, suggesting that other factor(s) within the plasma may mediate the conversion of the oligomeric pre-pore to a functional pore (*Galinier et al., 2013*). Although the functional mechanisms of these factors are not thoroughly understood, studies suggest that these factors function as key determinants in the final outcome of *S. mansoni* challenge of *B. glabrata* (*Galinier et al., 2013*; *Moné et al., 2010*).

While studies have implicated *Bg*FREP3 in resistance to *S. mansoni*, the underlying mechanism of *Bg*FREP3 function; how it binds to *S. mansoni* sporocyst surfaces, and then how recognition is translated into haemocyte engagement, activation, and ultimately parasite encapsulation, is still unknown (*Hanington et al., 2010*). Here, we report an association between *Bg*FREP3, *Bg*TEP1, Biomphalysin (UniProtKB/TrEMBL: A0A182YTN9 and A0A182YTZ4) and *Bg*FREP2 (UniProtKB/TrEMBL: A0A2C9L9F5). *Bg*FREP3 associates with *Bg*TEP1 and Biomphalysin in both M-line and BS-90 strains, but only in BS-90 strain uniquely interacts with *Bg*FREP2 and other versions of *Bg*FREP3 (a variant of *Bg*FREP3.3), providing us with an important insight into why BS-90 strain is refractory to *S. mansoni* infection. In this study, we demonstrate that *Bg*FREP3 binds to *S. mansoni* sporocysts without any other soluble plasma factors, yet binding of *Bg*FREP2 relies on *Bg*TEP1. However, *Bg*FREP3 still interacts with *Bg*TEP1 to form unique immune complexes, which significantly enhance the ability of

haemocytes and plasma from *S. mansoni*-susceptible *B. glabrata* (M-line) to kill *S. mansoni* sporocysts. A more striking finding is that the combination of *Bg*FREP3, *Bg*TEP1 and *Bg*FREP2 renders M-line haemocytes capable of killing *S. mansoni* sporocysts at nearly the same level as *S. mansoni*-resistant BS-90 haemocytes. Reactive oxygen species (ROS) are shown to play an important role during this haemocyte-mediated killing of *S. mansoni* sporocysts. These results provide insight into how the numerous previously characterized immune factors known to be important in the anti-*S. mansoni* immune response to *B. glabrata* are acting in concert to defend the snail host.

## Results

### *Bg*FREP3 interacts with *Bg*TEP1, Biomphalysin, *Bg*FREP2 and other *Bg*FREP3 variants in snail plasma

To identify the factors in snail plasma which interact with *Bg*FREP3, we produced recombinant *Bg*FREP3 (r*Bg*FREP3, GenBank: AY028461.1, M-line strain) using an insect expression system (*Figure 1A* and *Figure 1—source data 1*) and performed a series of pull-down experiments. We initiated our investigations with *Bg*FREP3 because we observed that the cell-free plasma of BS-90 snails contained more *Bg*FREP3 than does M-line snail plasma (*Figure 1B*). Among the proteins identified using liquid chromatography-tandem mass spectrometry (LC-MS/MS) in the eluent of these pull-down experiments was *Bg*TEP1 (*Figure 1C*). We performed four replicate r*Bg*FREP3 pull-down experiments, each yielded a band at approximately 200 kDa, which corresponded to the full-length *Bg*TEP1, which was identified from both M-line and BS-90 strains (*Figure 1C*). In three instances, we excised the bands from SDS-PAGE gels for mass spectrometry identification, this showed twice that *Bg*TEP1 was pulled down by r*Bg*FREP3 in M-line plasma, while this was shown once in both M-line and BS-90 plasma (*Figure 1D*). From the LC-MS/MS analysis we obtained a total of 17 unique peptides, 16 peptides were detected from M-line strain, four peptides were from BS-90 strain, and three peptides were present in both BS-90 and M-line strains (Table. S2 and *Figure 1D*). While all peptides mapped to *Bg*TEP, there was uncertainty as to which specific *Bg*TEP proteins are present. Twelve peptides mapped to *Bg*TEP (GenBank: ACL00841.1); 15 peptides mapped to *Bg*TEP1.1 (GenBank: ADE45332.1, from a Brazilian strain of *B. glabrata*), *Bg*TEP1.2 (GenBank: ADE45339.1, from a Brazilian strain of *B. glabrata*), *Bg*TEP1.3 (GenBank: ADE45340.1, from a Brazilian strain of *B. glabrata*), *Bg*TEP1.4 (GenBank: ADE45341.1, from a Brazilian strain of *B. glabrata*) and *Bg*TEP1.5 (GenBank: ADE45333.1, from a Brazilian strain of *B. glabrata*), however, there was overlap between some of these peptides, which mapped to all known *Bg*TEP sequences (*Figure 1—source data 2*). The *Bg*TEP identified by MS here displayed a relatively large difference from *Bg*TEP (ACL00841.1), but could not be distinguished from other *Bg*TEP1 sequences (*Figure 1—source data 2*), so we termed it *Bg*TEP1.

The full-length sequences of *Bg*TEP1 variants usually contain 1445 or 1446 amino acid residues (*Figure 1—source data 2*). The peptides identified by LC-MS/MS covered 16.3% of the full-length sequence (*Figure 1—source data 2*). The identified peptides were distributed over the full-length *Bg*TEP1, with the peptides identified from the BS-90 strain being concentrated in the N-terminal region (*Figure 1D*). Background transcript abundance of *Bg*TEP1 detected by qRT-PCR in M-line and BS-90 snails, suggested that there was no significant difference (p>0.05) in baseline *Bg*TEP1 expression between the two strains (*Figure 1—figure supplement 1*). These results indicate that the interaction of *Bg*FREP3 with *Bg*TEP1 doesn't require the involvement of any parasite components.

Recombinant *Bg*TEP1 (r*Bg*TEP1, GenBank: HM003907.1, Brazilian strain *B. glabrata*) was also used to perform a series of pull-down experiments (*Figure 1A and E*). Biomphalysin (UniProtKB/TrEMBL: A0A182YTN9 and A0A182YTZ4) was identified in both M-line and BS-90 plasma from r*Bg*TEP1 pull-down experiments and was also present in the pull-down studies using r*Bg*FREP3 (*Figure 1C and E*). Analysis suggested that Biomphalysin-A0A182YTN9 and Biomphalysin-A0A182YTZ4 amino acid sequences were 94% and 99% identical to the Biomphalysin (GenBank: AGG38744.1) previously reported (*Galinier et al., 2013*) (*Figure 1—source data 3A,B*). The 13 peptides identified by LC-MS/MS covered 28.3% of Biomphalysin (A0A182YTN9) amino acid sequence (*Figure 1F*, *Supplementary file 2* and *Figure 1—source data 3C*). These peptides were evenly distributed throughout the Biomphalysin-A0A182YTN9 protein (*Figure 1F* and *Figure 1—source data 3*). The theoretical molecular weight of Biomphalysin is ~64 kDa. Peptides identified from protein

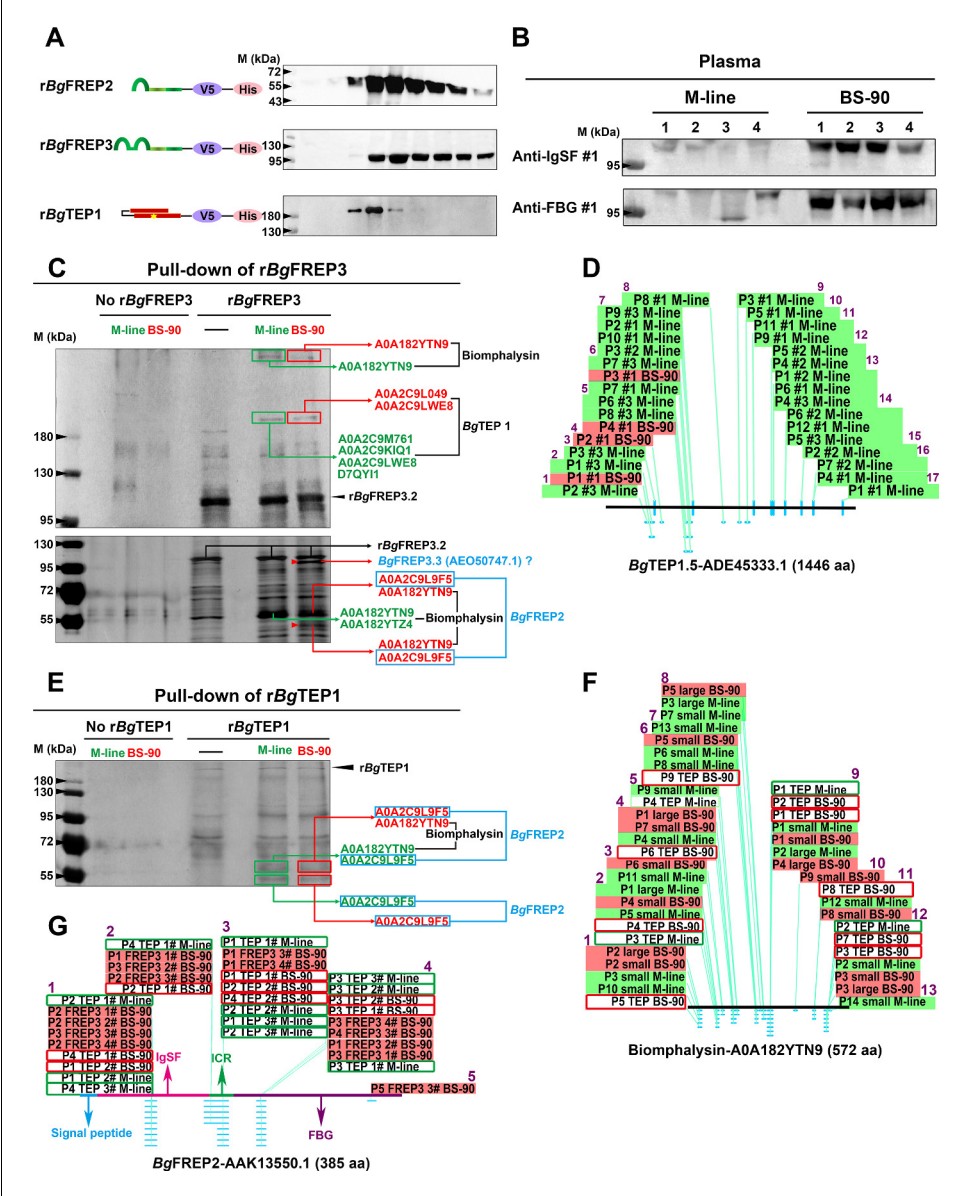

**Figure 1.** *Bg*FREP3, which had special high abundance in BS-90 plasma, interacted with *Bg*TEP1 and Biomphalysin derived from both M-line and BS-90 plasma, while uniquely associated with *Bg*FREP2 and a *Bg*FREP3 variant from BS-90 plasma. (**A**) Schematic diagram and Western blot analysis of recombinant proteins, the ends of which are coupled to the tags V5 and ploy-His for protein immunoassay and purification operations. The yellow star in the schematic structure of r*Bg*TEP1 represents the thioester motif. (**B**) The background expression of *Bg*FREP3 in BS-90 snail plasma was significantly higher than that in M-line plasma. The haemolymph of four snails from each M-line and BS-90 strain was independently extracted, and haemocytes were removed by centrifugation to obtain plasma samples. After the plasma of each snail was quantified, the same amount of plasma was used for SDS-PAGE electrophoresis and Western blot analysis. Anti-IgSF #1 and anti-FBG #1 antibodies of *Bg*FREP3 were used as primary antibodies. (**C**) The r*Bg*FREP3 associated with *Bg*TEP1 and Biomphalysin in both M-line and BS-90 plasma, but exclusively interacted with a *Bg*FREP3 variant and *Bg*FREP2 in BS-90 plasma. The eluent of r*Bg*FREP3 pull-downs was collected and separated on 6% (upper panel) and 10% (lower panel) SDS-PAGE gels before silver staining. (**E**) The r*Bg*TEP1 interacted with Biomphalysin and *Bg*FREP2 in both M-line and BS-90 plasma. The eluent of pull-down experiments with r*Bg*TEP1was separated on 10% SDS-PAGE gels. In the pull-down experiments with r*Bg*FREP3 (**C**) and r*Bg*TEP1 (**E**), cell-free plasma from M-line (green) or BS-90 (red) *B. glabrata* snails was incubated with Sf9 cell lysates expressing r*Bg*FREP3 or concentrated medium containing r*Bg*TEP1 together (interactome experiments, last two lanes) or alone (controls, first three lanes). Bands that differ between control and interactome experiments were cut, proteins submitted to tryptic digest and analyzed by LC-MS/MS for identification. In interactome experiments, bands that differ between M-line and BS-90 lanes were marked with red arrows. The results of mass spectrometry analysis are shown in the figure. Proteins (*Bg*TEP1 and Biomphalysin) represented by black font interacted with r*Bg*FREP3 in both M-line and BS-90 plasma. Proteins (*Bg*FREP2 and *Bg*FREP3.3) represented by blue font specially interacted with r*Bg*FREP3 only in BS-90. (**D, F and G**) Distribution of peptides identified by LC-MS/MS on *Bg*TEP1.5 (ADE45333.1), Biomphalysin (A0A182YTN9) and *Bg*FREP2 (AAK13550.1). The text on the panels describes the peptides identified from different Pull-down

*Figure 1 continued on next page*

*Figure 1 continued*

experiments, and most peptides were identified multiple times. The text with the green background describes peptides identified from the plasma of M-line strain in the r*Bg*FREP3 pull-down experiments, and corresponding text with the red background represents the peptides from the plasma of BS-90 strain. Similarly, the text in the green frame box represents the peptides from the plasma of the M-line strain in the r*Bg*TEP1 pull-down experiments, and corresponding text in the red frame box represents the peptides from the plasma of the BS-90 strain. The horizontal lines in panels D, F and G represent the full-length amino acid sequences of *Bg*TEP1.5, Biomphalysin and *Bg*FREP2. The light blue dots and dashes indicate the location of the peptides. The purple Arabic numerals represent the number of identified peptide fragments that differ from each other.

The online version of this article includes the following source data and figure supplement(s) for figure 1:

**Source data 1.** Nucleotide sequence for cloning r*Bg*FREP3.

**Source data 2.** The peptides identified by LC-MS/MS were distributed over the full-length *Bg*TEP, but could not be used to distinguish which *Bg*TEP variant was present.

**Source data 3.** The two Biomphalysin variants (UniProtKB/TrEMBL: A0A182YTN9 and A0A182YTZ4) identified by LC-MS/MS shared high sequence identities but were different from the previously published Biomphalysin (GenBank: AGG38744.1).

**Source data 4.** The identified protein (UniProtKB/TrEMBL: A0A2C9L9F5) represented a variant of *Bg*FREP2 members that was distinct from other *Bg*FREP members, but was indistinguishable from other *Bg*FREP2 variants.

**Source data 5.** Alignment of multiple *Bg*FREP3 amino acid sequences and distribution of identified peptides.

**Figure supplement 1.** Normalized transcript abundance of *Bg*TEP1 in unchallenged M-line (black bar, n = 5) and BS-90 (gray bar, n = 5) snails indicating a lack of significant difference (p>0.05) between the two strains.

pull-down that matched to Biomphalysin appeared at three different molecular weights, one at >400 kDa (*Figure 1C*), the second site at roughly 60 kDa (*Figure 1C and E*), which is in agreement with the theoretical molecular weight, and the third at around 55 kDa (*Figure 1C*), which might represent the size after proteolysis (*Galinier et al., 2013*).

Comparing r*Bg*FREP3 pull-down results between M-line and BS-90 *B. glabrata* identified two unique proteins in the BS-90 lane (*Figure 1C*). One of these proteins was *Bg*FREP2 (UniProtKB/TrEMBL: A0A2C9L9F5) (*Figure 1C and E*, *Figure 1—source data 4A*). Five peptides were identified by LC-MS/MS analysis and covered 16.6% of *Bg*FREP2 precursor (GenBank: AAK13550.1) amino acid sequence and were located throughout the IgSF and FBG of the protein (*Figure 1G*, *Supplementary file 2* and *Figure 1—source data 4*). Three of the five identified peptides specifically belonged to *Bg*FREP2 precursor (AAK13550.1), thus, the identified *Bg*FREP2-A0A2C9L9F5 appears to represent a specific *Bg*FREP2 variant that is distinct from other *Bg*FREP members that have been published (*Figure 1—source data 4B*), but not distinguishable from other *Bg*FREP2 members (*Figure 1—source data 4C,D*). BLAST analysis indicated that *Bg*FREP2-A0A2C9L9F5 had the highest identity (96.62%) to FREP A precursor (GenBank: NP_001298226.1) (*Figure 1—source data 4C*), which was constitutively expressed solely in haemocytes of the BS-90 resistant strain (*Dinguirard et al., 2018*).

Only *Bg*FREP2 was identified, at ~55 kDa, from the BS-90 strain plasma when r*Bg*FREP3 was used as the bait protein in pull-down experiments (*Figure 1C*). This contrasted with our other data in which *Bg*FREP2 was identified from plasma of both M-line and BS-90 strains when r*Bg*TEP1was used as bait in pull-down experiments (*Figure 1E*). *Bg*FREP2 was identified by LC-MS/MS from two distinct molecular weight bands in the BS-90 lane of the r*Bg*FREP3 study and the r*Bg*TEP1 study. One band, at ~60 kDa also included peptides that mapped to Biomphalysin while the other band identified at ~55 kDa, contained peptides for only *Bg*FREP2 (*Figure 1C and E*). Thus, all of the *Bg*FREP2 peptides were identified from bands that were of greater molecular weight than the predicted *Bg*FREP2, which was ~43 kDa. However, this was consistent with the observed molecular weight of *Bg*FREP2 by *Moné et al. (2010)*, and could be explained by post-translational modifications such as glycosylation (*Adema et al., 1997*; *Moné et al., 2010*; *Zhang et al., 2008*).

Another unique protein identified from the BS-90 lane in the r*Bg*FREP3 pull-down experiments was a variant of *Bg*FREP3 (*Figure 1C*). Of the nine peptides identified by LC-MS/MS, seven specifically matched *Bg*FREP3.3 (Genbank: AEO50747.1), the other two matched other members of *Bg*FREP3 family: *Bg*FREP3.3 (GenBank: AAO59915.1), *Bg*FREP3.2 (GenBank: AEO50746.1), *Bg*FREP3.2 (GenBank: AAK28656.1) and *Bg*MFREP3 (GenBank: AAK13548.1), suggesting that this protein was likely a FREP3 variant (*Figure 1—source data 5* and *Supplementary file 2*). The seven peptides that specifically belonged to *Bg*FREP3.3 (AEO50747.1) covered 15.6% of the amino acid sequence length and mainly concentrated in the ICR (five peptides), and one peptide in each of the

IgSF2 domain and FBG domain (*Figure 1—source data 5*). These results suggest that *Bg*FREP3, *Bg*TEP1, Biomphalysin and *Bg*FREP2 associate with each other in *B. glabrata* plasma, with M-line and BS-90 strains displaying differing interactomes. Our data indicates that *Bg*FREP3 specially interacts with *Bg*FREP2 and another unique *Bg*FREP3 variant only in BS-90 *B. glabrata*. These results provide us with a new perspective to explore why M-line and BS-90 strains have differing compatibility phenotypes to *S. mansoni* infection.

## *Bg*FREP3 independently recognizes and binds to primary sporocysts of *S. mansoni*

Our data suggests a close and complex interaction between *Bg*FREP3, *Bg*TEP1, Biomphalysin and *Bg*FREP2. These factors have each independently been reported to be involved in the recognition of *S. mansoni* sporocysts (*Adema et al., 1997*; *Galinier et al., 2013*; *Hanington et al., 2010*; *Hanington et al., 2012*; *Moné et al., 2010*; *Portet et al., 2018*; *Wu et al., 2017*). However, only Biomphalysin is known to directly bind to the surface of *S. mansoni* sporocysts without the aid of any other plasma factors (*Galinier et al., 2013*). The mechanism by which *Bg*FREP3, *Bg*TEP1 and *Bg*FREP2 bind to primary sporocysts of *S. mansoni* is still not clear. To explore this issue, we produced r*Bg*FREP3, r*Bg*TEP1 and r*Bg*FREP2 (GenBank: AY012700.1, M-line strain) to observe whether they associate with primary sporocysts of *S. mansoni* (*Figure 1A*). Immunocytochemistry clearly showed that r*Bg*FREP3 and combination of r*Bg*FREP3 and r*Bg*TEP1 were able to interact with the outer tegument of *S. mansoni* sporocysts (*Figure 2*), while r*Bg*TEP1 and r*Bg*FREP2 alone did not have such capabilities (*Figure 2* and *Figure 2—figure supplement 1*). However, incubation of sporocysts with pre-combined r*Bg*FREP2 and r*Bg*TEP1 yielded a signal, suggesting that these two factors required complex formation prior to being able to associate with *S. mansoni* sporocysts (*Figure 2* and *Figure 2—figure supplement 1*). These results indicate the mechanisms by which *Bg*FREP3 and *Bg*FREP2 recognize *S. mansoni* sporocysts are different, that is, *Bg*FREP3 does not require *Bg*TEP1 in order to associate with targets on the *S. mansoni* sporocyst surface while *Bg*FREP2 relies on *Bg*TEP1 to bind to *S. mansoni* sporocysts.

## *Bg*FREP3 and *Bg*TEP1 form an immune complex

To further investigate whether *Bg*FREP3 and *Bg*TEP1 form a complex, we carried out a set of immunoblotting experiments with recombinant proteins. As expected, r*Bg*FREP3 did form a complex with r*Bg*TEP1 (*Figure 3A*). The complex appeared in a region of high molecular weight (>460 kDa), and the signal decreased gradually with the decrease of r*Bg*TEP1:r*Bg*FREP3 ratios in a dose-dependent manner (*Figure 3A*). As shown in *Figure 3A*, r*Bg*FREP3 was found as a high molecular weight multimer in a non-denatured state. The molecular weights of the r*Bg*FREP3 multimer and the r*Bg*FREP3-r*Bg*TEP1 complex were both >460 kDa, thus, it was difficult to distinguish them using Western blot. To confirm the interaction between r*Bg*FREP3 and r*Bg*TEP1, a far-Western blotting approach, in which decreasing amounts of r*Bg*TEP1 were separated by SDS-PAGE, and transferred to a nitrocellulose membrane, and then detected by using biotinylated r*Bg*FREP3 was performed. A band was detected in the lane with 2 μg of r*Bg*TEP1, confirming the interaction between r*Bg*FREP3 and r*Bg*TEP1 (*Figure 3B*).

*Bg*FREP2 and *Bg*TEP1 have been shown to form immune complexes with *Sm*PoMucs derived from *S. mansoni* (*Moné et al., 2010*), suggesting that *Bg*FREP2 and *Bg*TEP1 associate with each other in snail plasma. To test this assumption, we performed a set of immunoblot assays using r*Bg*FREP2 and r*Bg*TEP1. The results of Western blot under non-denaturing conditions showed a band (>460 kDa) after mixing r*Bg*FREP2 and r*Bg*TEP1 (*Figure 3C* upper panel). The intensity of this band decreased on the Western blot in a dose dependent manner along with the decrease of the r*Bg*TEP1:r*Bg*FREP2 ratios (*Figure 3C* upper panel). However, r*Bg*FREP2 did not exist as a multimer under non-denaturing conditions (*Figure 3C* upper panel), which was consistent with the findings of (*Zhang et al., 2008*). In addition, we further verified the interaction between r*Bg*FREP2 and r*Bg*TEP1 by Far-Western blot (*Figure 3C* lower panel).

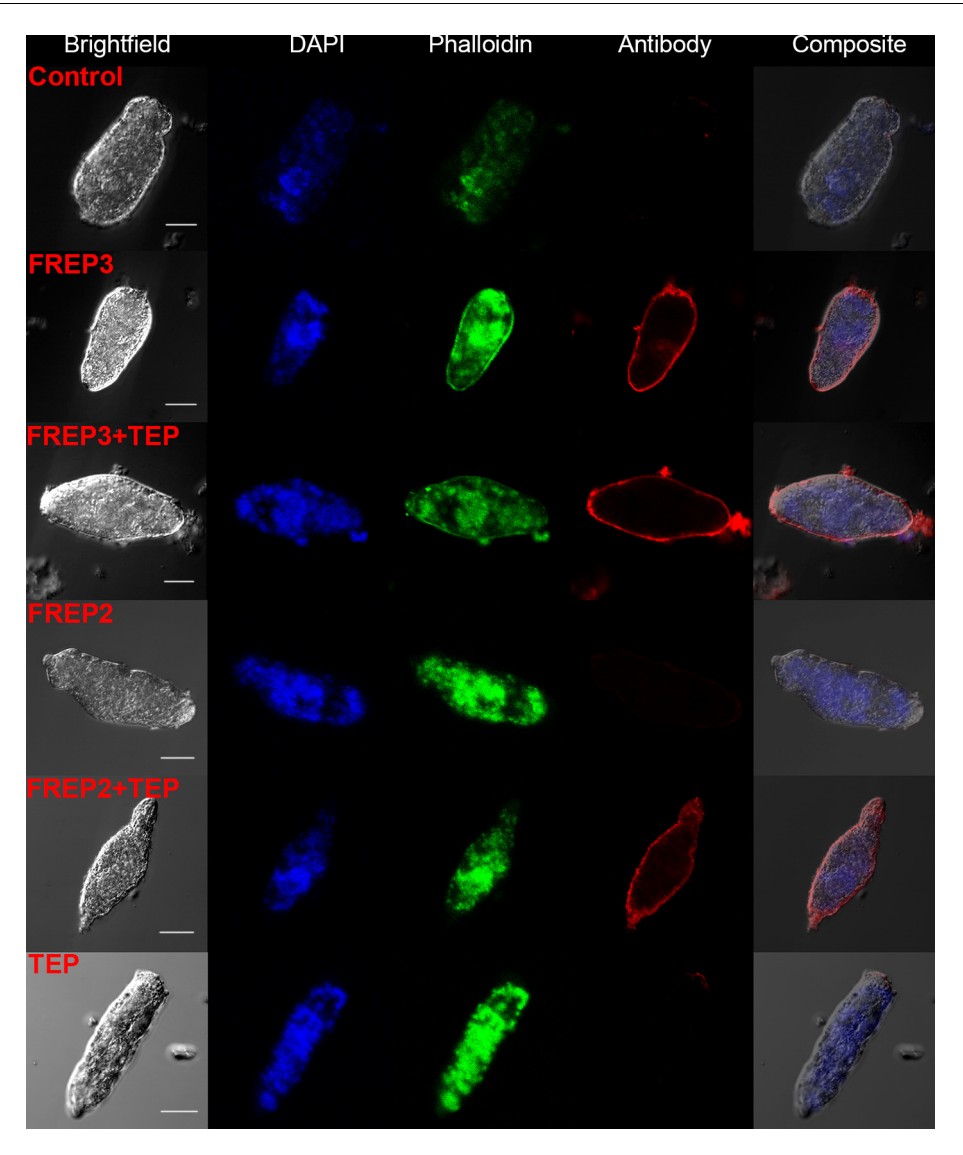

**Figure 2.** A combination of r*Bg*FREP2 + r*Bg*TEP1, and r*Bg*FREP3 in the presence and absence of r*Bg*TEP1, are capable of binding to the surface of *S. mansoni* sporocysts. Fixed sporocysts were incubated with a vehicle control or recombinant proteins (r*Bg*FREP3, r*Bg*FREP2, and r*Bg*TEP1 alone) and combinations of r*Bg*FREP2+r*Bg*TEP1 or r*Bg*FREP3+r*Bg*TEP, followed by immunostaining with anti-V5 primary IgG and Alexa Fluor 633 goat anti-mouse secondary antibody. White bar in the bright field column represents 25 μm.

The online version of this article includes the following figure supplement(s) for figure 2:

**Figure supplement 1.** Recombinant proteins r*Bg*FREP3, r*Bg*FREP2 and r*Bg*TEP1 bound to the outer tegument of live *S. mansoni* sporocysts.

## Association of *Bg*FREP3 and *Bg*TEP1 facilitates recognition and killing of *S. mansoni* sporocysts

*Bg*FREP3 does not rely on *Bg*TEP1 to recognize and bind to *S. mansoni*-associated proteins, but does associate with *Bg*TEP1, because of this, we hypothesize that *Bg*TEP1 plays a role in the downstream immune response triggered by *Bg*FREP3 recognition of *S. mansoni* sporocysts. We treated primary sporocysts with haemocytes or plasma from BS-90 and M-line *B. glabrata* snails with r*Bg*FREP3, r*Bg*TEP1 and a combination of the two in vitro.

Plasma (haemoglobin-low/free) from the resistant BS-90 strain of *B. glabrata* snails was naturally able to kill 67% (SEM 3.7%; n = 10) of *S. mansoni* sporocysts by 48 hr post incubation, while plasma

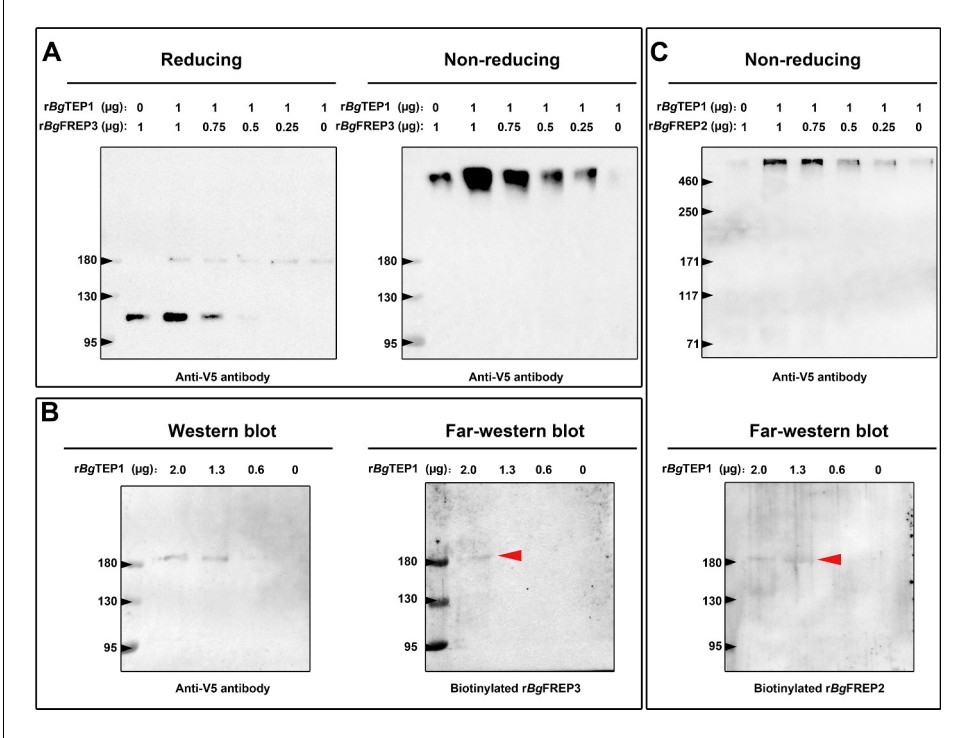

**Figure 3.** Both rBgFREP3 and rBgFREP2 interacted with rBgTEP1 to form complexes. (**A**) Purified rBgTEP1 was incubated with rBgFREP3 in the indicated ratios (wt/wt in μg) for 2 hr at room temperature. After the incubation, Western blot analysis was carried out under denaturing (heating with reducing agent, the right) and non-denaturing (no heating, no reducing agent, the left) conditions, respectively. (**B**) Decreasing amounts of rBgTEP1 were blotted onto the membrane and then subjected to standard Western blot (the left) and far-Western blot (the right) analysis. (**C**) The interaction between rBgFREP2 and rBgTEP1. Western blot analysis under non-denaturing (the upper) and far-Western blot analysis (the below) was carried out, respectively. In the far-Western blot, the biotinylated rBgFREP3 and rBgFREP2 acts as a primary antibody and the secondary antibody is a streptavidin-HRP; the red arrow indicates the band that symbolizes the interaction between BgTEP1 and rBgFREP3 or rBgFREP2.

from the susceptible M-line strain, which killed 20% (SEM 4.5%; n = 10), was not significantly more effective than the medium control in which 13% (SEM 2.1%; n = 10) were killed at 48 hr (*Figure 4A*). The addition of rBgFREP3 (36% [SEM 5.0%; n = 10]) and rBgTEP1 (32% [SEM 3.3%; n = 10]) independently did not significantly enhanced the ability of M-line plasma to kill *S. mansoni* sporocysts by 48 hr (*Figure 4A*). However, the combined addition of rBgFREP3 and rBgTEP1 significantly increased M-line-plasma-mediated killing of *S. mansoni* sporocysts by 48 hr post incubation (56% [SEM 4.8%; n = 10]) compared to the application of either recombinant protein alone. The combined addition of rBgFREP3 and rBgTEP1 rendered the capacity of M-line plasma able to kill *S. mansoni* sporocysts statistically insignificant from BS-90 plasma (p>0.05) (*Figure 4A*). Recombinant proteins, in the absence of *B. glabrata* plasma, did not display any capacity to kill *S. mansoni* sporocysts (*Figure 4— figure supplement 1A*). This suggests that rBgFREP3 and rBgTEP1 activate one or more factors in plasma to confer the ability to kill sporocysts. Combined with the results of our rBgFREP3 and rBgTEP1 pull-down experiments, we speculate that Biomphalysin plays an important role in this process. However, an alternative mechanism underpinning plasma-mediated killing may be elevated circulating levels of ROS (e.g., $H_2O_2$) naturally occurring in the plasma of BS-90 compared to M-line *B. glabrata*. To address this possibility, catalase (a ROS scavenger) was pre-incubated with BS-90 plasma and M-line plasma. Catalase did not affect the *S. mansoni* sporocyst killing capacity of BS-90 plasma nor did it influence M-line plasma incubated with rBgFREP3 and rBgTEP1 (*Figure 4—figure supplement 1B*), suggesting that other factors besides ROS were activated by rBgFREP3 and rBgTEP1 to mediate sporocyst killing.

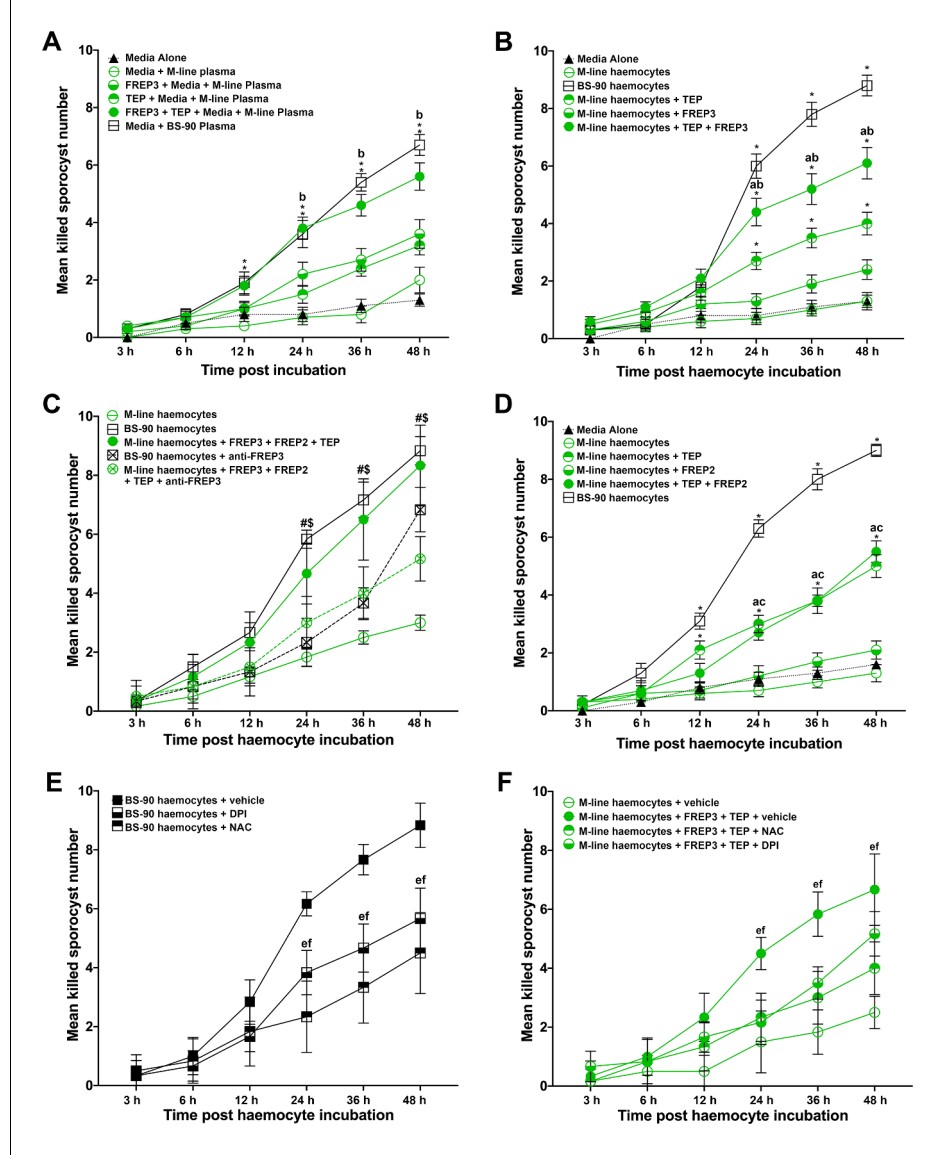

**Figure 4.** Association of r*Bg*FREP3, r*Bg*TEP1 and r*Bg*FREP2 increased the ability of M-line snails to kill *S. mansoni* sporocysts. Effects of r*Bg*FREP3, r*Bg*TEP1 and r*Bg*FREP3-r*Bg*TEP1 complex on killing of *S. mansoni* sporocysts by M-line plasma (**A**) and haemocytes (**B**). (**C**) Before being exposed to sporocysts, M-line haemocytes were pre-incubated with r*Bg*FREP3-r*Bg*TEP1-r*Bg*FREP2 combination in presence or in absence of anti-*Bg*FREP3 antibodies (abrogation treatment). BS-90 haemocytes were incubated with anti-*Bg*FREP3 antibodies. (**D**) The effect of r*Bg*FREP2, r*Bg*TEP1 and r*Bg*FREP2-r*Bg*TEP1 complex on the destruction of sporocysts by M-line haemocytes. (**E** and **F**) The role of ROS in killing of *S. mansoni* sporocysts by haemocytes from *B. glabrata* snails. (**E**) The addition of ROS inhibitors NAC and DPI abolished the ability of BS-90 haemocytes to destroy sporocysts. (**F**) Killing sporocysts of M-line haemocytes rendered by r*Bg*FREP3-r*Bg*TEP1 complex was annulled by pre-incubation with ROS inhibitors NAC and DPI. Statistically significant difference symbols in **A**, **B** and **D**: asterisk (*) represents comparison with M-line haemocytes or plasma, *$p<0.05$, **$p<0.01$; a represents comparison with BS-90 haemocytes or plasma, $p<0.05$; b represents comparison with both single treatment (r*Bg*TEP1/r*Bg*FREP3 or r*Bg*FREP2), $p<0.05$; c represents comparison with only *Bg*FREPs treatment, $p<0.05$; bars represent SEM, n = 10. Statistically significant difference symbols in **C**: # represents comparison BS-90 haemocytes with BS-90 haemocytes +anti-r*Bg*FREP3 antibodies, $p<0.05$; $ represents comparison M-line haemocytes+r*Bg*FREP3+r*Bg*FREP2+r*Bg*TEP1 with M-line haemocytes+r*Bg*FREP3+r*Bg*FREP2+r*Bg*TEP1+anti-r*Bg*FREP3 antibodies, $p<0.05$; bars represent SEM, n = 10. Statistically significant difference symbols in **E** and **F** e represents comparison NAC treatment with BS-90 haemocytes or M-line haemocytes+r*Bg*FREP3+r*Bg*TEP1, $p<0.05$; f represents comparison DPI treatment with BS-90 haemocytes or M-line haemocytes+r*Bg*FREP3+r*Bg*TEP1, $p<0.05$; bars represent SEM, n = 6.

*Figure 4 continued on next page*

*Figure 4 continued*
The online version of this article includes the following source data and figure supplement(s) for figure 4:
**Source data 1.** *S. mansoni* sprorcyst killing assay raw data.
**Figure supplement 1.** Controls of in vitro *S. mansoni* sporocyst killing assays.

Haemocytes derived from BS-90 strain *B. glabrata* were innately capable of destroying *S. mansoni* sporocysts, killing 88% (SEM 3.6%; n = 10) of sporocysts after 48 hr of incubation in vitro (*Figure 4B*). The basal ability of M-line haemocytes to kill sporocysts (13% [SEM 3%; n = 10]) was not significantly different from the medium-alone control (13% [SEM 2.1%; n = 10]), and was only slightly increased by adding rBgFREP3 (24% [SEM 3.4%; n = 10]) at 48 hr (*Figure 4B*). Addition of rBgTEP1 did significantly increase sporocyst killing by M-line haemocytes to 40% (SEM 3.9%; n = 10) (*Figure 4B*), however, this killing response was significantly enhanced to 61% (SEM 5.5%; n = 10) when rBgFREP3 and rBgTEP1 were incubated with M-line haemocytes together (*Figure 4B*). However, the combination of rBgFREP3 and rBgTEP1 did not convey a killing capacity to M-line snails that paralleled BS-90 haemocytes (*Figure 4B*). This suggests that there is yet another factor(s) affecting this process, causing haemocytes from BS-90 and M-line to differ in their ability to destroy sporocysts. In our rBgFREP3 pull-down experiments, BgFREP2 was one of the two proteins uniquely identified as associating with rBgFREP3 in the plasma of BS-90 *B. glabrata* (*Figure 1C*). Incubation of rBgFREP2 in combination with rBgFREP3 and rBgTEP1 enabled M-line haemocytes to kill *S. mansoni* sporocysts (83% [SEM 4.8%; n = 6]) to statistically indistinguishable (p>0.05) levels as BS-90 haemocytes (88% [SEM 4.7%; n = 6]) (*Figure 4C*). To verify whether the rBgFREP2-rBgTEP1 complex dramatically enhanced the ability of M-line haemocytes to killing *S. mansoni* sporocysts above either factor independently, as did the rBgFREP3-rBgTEP1 complex, a similar experiment was performed, showing that rBgFREP2-rBgTEP1 did not play such a role (*Figure 4D*). Independent treatments of rBgFREP2 (21% [SEM 3.1%; n = 10]) and rBgTEP1 served as controls (*Figure 4—figure supplement 1C*), demonstrating that BgFREP2 alone did not facilitate sporocyst killing above control, whereas rBgTEP1 killing (50% [SEM 3.9%; n = 10]) was indistinguishable (p>0.05) from rBgFREP2 and rBgTEP1 combined (55% [SEM 3.7%; n = 10]) (*Figure 4D*).

These results made us speculate that BgFREP3 was the determining factor controlling immune engagement with sporocysts and subsequent killing. Addition of three anti-rBgFREP3 antibodies to the mixture of rBgFREP3, rBgTEP1 and rBgFREP2 significantly abrogated the increase in the ability of M-line haemocytes to kill sporocysts (51% [SEM 3.1%; n = 6] compared to 83% [SEM 4.8%; n = 6]) (*Figure 4C*). Pre-incubation of BS-90 haemocytes with the anti-rBgFREP3 antibodies also significantly reduced their ability to destroy sporocysts (68% [SEM 3.1%; n = 6] compared to 88% [SEM 4.7%; n = 6]) (*Figure 4C*). Combined pre-immunized IgG purified from serum from the same rabbits from which the antibodies were produced was used as a negative control, demonstrating that it did not significantly affect the killing *S. mansoni* sporocysts by M-line (51% [SEM 3%; n = 6]) and BS-90 haemocytes (90% [SEM 3.7%; n = 6]) (*Figure 4—figure supplement 1D*). These results suggest that rBgFREP3 plays a central role in initiating the snail haemocyte-mediated anti-*S. mansoni* immune responses.

Finally, we hypothesize that haemocyte-mediated killing likely involves the production of ROS, which is crucial to the clearance of *S. mansoni* sporocysts (*Hahn et al., 2000*; *Hahn et al., 2001*). To investigate the role of ROS in the combined ability of rBgFREP3 and rBgTEP1 to enhance the ability of M-line haemocytes to kill *S. mansoni* sporocysts, ROS inhibitors N-acetyl-L-cysteine (NAC), Diphenyleneiodonium (DPI) and catalase were applied to the sporocyst killing assays. Co-incubation of BS-90 haemocytes with NAC, DPI and catalase significantly reduced their ability to kill *S. mansoni* sporocysts (45% [SEM 5.6%; n = 6], 57% [SEM 4.2%; n = 6] and 60% [SEM 5.9%; n = 6] respectively) compared to BS-90 haemocytes in the vehicle control (89% [SEM 3.1%; n = 6]) by 48 hr post incubation (*Figure 4E*, *Figure 4—figure supplement 1A*). The ability of M-line haemocytes primed by the rBgFREP3-rBgTEP1 complex to destroy sporocysts (67% [SEM 4.9%; n = 6]) was also significantly abrogated by the addition of NAC (40% [SEM 3.7%; n = 6]) or DPI (52% [SEM 3.1%; n = 6]) (*Figure 4F*).

## Discussion

In 1984, Granath and Yoshino successfully achieved passive transfer of resistance to *S. mansoni* from a refractory *B. glabrata* strain to susceptible snails via injection of cell-free haemolymph (plasma). Similar passive transfer of resistance had previously been shown using an in vitro model (*Bayne et al., 1980a*; *Bayne et al., 1980b*; *Loker and Bayne, 1982*). Although the oxidation of hae-moglobin, which is the main protein component (97% of total protein) in *B. glabrata* plasma was later found to be toxic to *S. mansoni* sporocysts in long-term cultures (*Bender et al., 2002*), and thus likely explained the passive transfer study results, further studies suggest that specific anti-para-sitic factors must be present at or above a certain threshold level in *S. mansoni*-resistant snail plasma to stimulate a haemocyte-mediated destruction of sporocysts within 24–48 hr post-infection (*Dinguirard et al., 2018*). Although many important immune factors have been identified from *B. glabrata* plasma/haemocytes, such as *Bg*FREPs (*Adema et al., 1997*), *Bg*TEP1 (*Moné et al., 2010*), Biomphalysin (*Galinier et al., 2013*), Toll-like receptors (*Bg*TLRs) (*Pila et al., 2016b*), granulin (*Bg*GRN) (*Pila et al., 2016a*) and macrophage migration inhibitory factor (*Bg*MIF) (*Baeza Garcia et al., 2010*), it is still not known which factors ultimately underpin *B. glabrata* immunity to *S. mansoni*. Here, we provide insight into the broader interactions that take place between *B. glabrata* immune factors that leads to development of an *S. mansoni*-resistant phenotype.

Previous studies have shown that *Bg*FREPs are capable of binding to digenean trematode sporo-cysts and are up-regulated in response to larval infection (*Adema et al., 1997*; *Wu et al., 2017*; *Zhang et al., 2001*). *Bg*FREP3 has been shown to be a central part of the anti-*S. mansoni* immune response (*Hanington et al., 2010*). *Bg*FREP3 is up-regulated based on three characteristics (age, strain, and acquired resistance) of resistance of *B. glabrata* to infection with *S. mansoni* or *Echinos-toma paraensei* (*Hanington et al., 2010*). Knock-down of *Bg*FREP3 in resistant snails using siRNA-mediated interference altered the phenotype of resistant *B. glabrata* by increasing susceptibility to *E. paraensei* infection (*Adema, 2015*; *Hanington et al., 2010*). Nevertheless, we have little knowl-edge of the underlying mechanism by which *Bg*FREP3 interacts with other snail plasma factors in order to signal and trigger an immune response after recognizing the pathogen. In order to explore this mechanism, we conducted a series of pull-down experiments that allowed us to discover the interaction between *Bg*TEP1, Biomphalysin and *Bg*FREP2 with *Bg*FREP3 (*Figure 1*).

Proteomic analysis of *B. glabrata* plasma revealed that *Bg*FREP3, *Bg*TEP1 and *Bg*FREP2 display affinity for *S. mansoni* sporocyst tegumental membrane proteins (*Portet et al., 2018*; *Wu et al., 2017*). *Bg*TEP1 was previously reported to play a role in *Bg*FREP2 recognition of *S. mansoni* sporo-cyst *Sm*PoMucs, (*Moné et al., 2010*; *Portet et al., 2018*). Our immunofluorescence results illustrate that *Bg*FREP3 independently recognizes *S. mansoni* sporocysts, whereas *Bg*FREP2 must be involved with the participation of *Bg*TEP1 (*Figure 2*). We speculate that this may be due to their structural dif-ferences: *Bg*FREP3 possesses two IgSF domains, yet *Bg*FREP2 has only one; *Bg*FREP3 exists as a multimer in its natural state (*Figure 3*), whereas *Bg*FREP2 does not (*Zhang et al., 2008*). Alterna-tively, it is possible that the targets they recognize are different, so the immunoaffinity reactions that occur are also fundamentally different. A recent report suggests that *Bg*FREPs are likely to form mul-timers through the coiled-coil region of the ICR. However, this has not been supported by experi-mental evidence and does not completely rule out the possibility that *Bg*FREP multimer formation is mediated by the FBG or IgSF domain (*Gorbushin, 2019*).

*Bg*TEP1 and *Bg*FREP3 form a complex without any *S. mansoni* molecules present (*Figure 3*). We do not yet know exactly how *Bg*TEP1 is activated during infection. It is possible that *Bg*FREP3 is sta-bilizing *Bg*TEP1 like the leucine-rich repeat immune protein (LRIM)/*Anopheles Plasmodium*-respon-sive Leucine-rich repeat protein 1 (APL1C) complex that has been found to stabilize TEP in *Anopheles gambiae,* while also guiding it to the sporocyst's surface (*Fraiture et al., 2009*). Addition-ally, our LC-MS/MS results seem to indicate that *Bg*FREP3 interacted with a full-length form of *Bg*TEP1, while previous work has only found the C-terminal portion of *Bg*TEP1 associating with *Sm*PoMucs (*Moné et al., 2010*). This may indicate that once brought in association with the parasite, *Bg*TEP1 becomes cleaved/activated. *Bg*FREP3 does not require *Bg*TEP1 to recognize pathogens, this suggests *Bg*TEP1 is more likely to play a role in the downstream immune response initiated by *Bg*FREP3. Following the addition of r*Bg*FREP3 and r*Bg*TEP1, plasma demonstrated an increased capacity to kill sporocysts compared to controls (*Figure 4A*). Toxicity of haemoglobin oxidation to *S. mansoni* sporocysts was ruled out as a possible mechanism underpinning sporocyst killing because

the *B. glabrata* plasma used in these tests had haemoglobin removed. These studies also demonstrated that recombinant *Bg*FREP3 and *Bg*TEP1 alone were not able to increase sporocyst killing, but showed that the addition of cell-free plasma with both recombinants was required, thereby implying the necessity of another factor(s). Based on our pull-down experiments, we suggest that this factor could be a Biomphalysin. Biomphalysin possesses many similarities to members of the β-PFT superfamily (*Howard and Buckley, 1985*; *MacKenzie et al., 1999*; *Wilmsen et al., 1992*; *Xu et al., 2014*), and has been shown to have cytolytic activity that is mediated by a plasma factor(s) that remains to be identified (*Galinier et al., 2013*). Our results suggest a scenario in which *Bg*FREP3 and/or *Bg*TEP1 could be the factors associating with Biomphalysin to promote plasma-mediated sporocyst killing. *Bg*FREP3 and/or *Bg*TEP1 might activate Biomphalysin in an unknown manner, assist in the oligomerization of Biomphalysin while it forms its heptameric channel, or mediate the conversion of the oligomeric pre-pore to a functional pore. Further investigation is warranted to illustrate the role of *Bg*FREP3 and *Bg*TEP1 in the activation of Biomphalysin.

Within 2–3 hr of *S. mansoni* entry into *B. glabrata*, haemocytes surround the developing sporocysts forming multi-layered cellular capsules that kill encapsulated larvae within 24–48 hr post-infection (*Dinguirard et al., 2018*; *Loker and Bayne, 1982*). This occurs in both susceptible and resistant snail strains and has been replicated in our in vitro functional studies, where only resistant snails can eventually destroy the sporocysts without external assistance. Dinguirard et al. conducted a comparative proteomic analysis on the responses of haemocytes participating in *S. mansoni* sporocyst encapsulation reactions from susceptible and resistant *B. glabrata* strains, showing a striking differences in proteins expressed, with susceptible snail haemocytes exhibiting extensive downregulation of protein expression and a lower level of constitutively expressed proteins involved immunity (e.g., *Bg*FREP2) and ROS production compared to resistant snails (*Dinguirard et al., 2018*). This could explain why BS-90 haemocytes alone can kill *S. mansoni* sporocysts under normal in vivo conditions. After challenge, the transcript abundance of numerous *Bg*FREPs increases by 3-fold or greater in BS-90 compared to M-line *B. glabrata* (*Adema et al., 1997*). Noteworthy in the context of our current study is that following challenge BS-90 snails with *S. mansoni*, transcript abundance for *Bg*FREP2 increases over 50-fold by 1 day post-exposure (*Dinguirard et al., 2018*; *Hertel et al., 2005*), *Bg*FREP3 increases around 4-fold by 1 day post-exposure (*Hanington et al., 2010*), and *Bg*TEP1 which increases by 2- to 3.5-fold from 6 to 24 hr post-exposure (*Portet et al., 2018*). These dramatic increases in the abundance of immune-relevant transcripts occurring over the first 12–24 hr post exposure to *S. mansoni* sporocysts in BS-90 snails aligns closely with our results that demonstrate sporocyst killing by BS-90 haemocytes rapidly increases after 12 hr post-exposure (*Figure 4*).

Numerous studies have demonstrated the production of ROS by haemocytes in *B. glabrata*, especially $H_2O_2$, which plays a vital role in anti-schistosome defense (*Galinier et al., 2013*; *Hahn et al., 2000*; *Hahn et al., 2001*). Haemocytes from *S. mansoni*-resistant snails produce significantly more ROS than susceptible snails (*Bender et al., 2005*; *Bender et al., 2007*; *Goodall et al., 2004*). In our study, ROS was critical in BS-90 haemocyte-mediated *S. mansoni* sporocyst killing, as this ability was significantly reduced after pre-incubation with the ROS inhibitors NAC and DPI (*Figure 4E*). However, according to the phenomenon of passively transferred resistance, the cytotoxic potential of susceptible haemocytes may not be fundamentally different from resistant snails (*Granath and Yoshino, 1984*). We infer that M-line plasma may lack some factors or may not achieve sufficient thresholds of certain factors to stimulate haemocytes to produce ROS. It is also possible that M-line haemocytes are lacking specific receptors to receive signals that upregulate ROS to effective levels able to kill *S. mansoni* sporocysts. Previously, our team identified a toll-like receptor (*Bg*TLR) from haemocyte surface of *B. glabrata* snails, which differed in abundance between M-line and BS-90 strains (*Pila et al., 2016b*). It has been repeatedly demonstrated in other organisms that activation of Toll-like receptors is associated with increased intracellular ROS (*Asehnoune et al., 2004*; *Tsung et al., 2007*; *Wong et al., 2009*).

Our results indicate that the *Bg*FREP3-*Bg*TEP1 complex plays a role in raising ROS levels in M-line haemocytes (*Figure 4F*). We found that BS-90 haemolymph contains more *Bg*FREP3 protein than M-line (*Figure 1B*). Moreover, the BS-90 haemocytes significantly increased the expression level of *Bg*FREP2 during encapsulating *S. mansoni* sporocysts compared with susceptible snail haemocytes (*Dinguirard et al., 2018*). These suggest that *Bg*FREP3 and *Bg*FREP2 may play a crucial role in inducing haemocytes which take part in encapsulating sporocysts to generate ROS. We suppose

that *Bg*TEP1 also plays an opsonin role in the *B. glabrata* snail haemolymph, which interacts with molecules on the surface of haemocytes, thereby promoting the phagocytosis of pathogens.

Our data suggests that there are fundamental differences between the *Bg*FREP3-mediated immune response between BS-90 and M-line *B. glabrata*. Protein Pull-down experiments yielded two obvious and unique protein bands that were present in BS-90 but not M-line pull-downs. We identified the bands to be a variant of *Bg*FREP3.3 and *Bg*FREP2 (*Figure 1C*), showing different immune factor interactomes between different strains. This suggests that *Bg*FREP3 variants and *Bg*FREP2 in BS-90 plasma are more prone to r*Bg*FREP3 binding, and the combination of different versions of *Bg*FREPs seems to play an important and unknown role in snail resistance. Both *Bg*FREP2 and *Bg*FREP3 are characterized by high diversity, our results imply that the versions of *Bg*FREP2 and *Bg*FREP3 in the two snail strains are different. After exposure to *S. mansoni*, the expression of *Bg*FREP2 in resistant snails is superior to that of susceptible snails (*Dinguirard et al., 2018*; *Gordy et al., 2015*; *Hertel et al., 2005*). This again suggests that *Bg*FREP2 is important in determining the outcome of the compatibility of *B. glabrata* against *S. mansoni*. Our study implies that *Bg*FREP3 seems to play an 'all or none' switching effect in determining *B.glabrata-S. mansoni* compatibility, while *Bg*FREP2 seems to play an additive role.

It has been decades since the discovery and initial characterization of *Bg*FREPs in the *B. glabrata* immune response to *S. mansoni*. While numerous studies since have implicated *Bg*FREP3 and *Bg*FREP2 along with a suite of other immune factors, as being important elements of the snail immune response to *S. mansoni*, how the response is coordinated has remained elusive. Our study presents a model in which many of the known determinants of snail resistance to *S. mansoni* work in concert to protect the snail host against infection. We map a mechanism for the interaction of *Bg*FREP3, *Bg*TEP1, *Bg*FREP2 and Biomphalysin with *S. mansoni* sporocysts in *B. glabrata* snail haemolymph that ultimately leads to sporocyst killing (*Figure 5*). Central to this response is *Bg*FREP3, which is able to recruit unique representatives of both *Bg*FREP2 and *Bg*FREP3 in *S. mansoni*-resistant BS-90 *B. glabrata*. With this new-found understanding of the *B. glabrata* immune response comes the ability to now undertake comprehensive assessments of how modifications to both the snail host and the parasite influence the underpinning immunological drivers of compatibility in this important model for studying human schistosomes.

## Materials and methods

### Live material and experimental treatments

M-line and BS-90 strain *B. glabrata* snails were used in this study. M-line and BS-90 strain *B. glabrata* snails, and *S. mansoni* were maintained at the University of Alberta as described previously (*Pila et al., 2016a*). NMRI strain of *S. mansoni* was obtained from infected Swiss-Webster mice provided by the NIH/NIAID Schistosomiasis Resource Center at the Biomedical Research Institute (*Cody et al., 2016*). All animal work observed ethical requirements and was approved by the Canadian Council of Animal Care and Use Committee (Biosciences) for the University of Alberta (AUP00000057).

### Recombinant *Bg*FREP3, *Bg*TEP1 and *Bg*FREP2 Synthesis and Purification

Recombinant *Bg*FREP3 (r*Bg*FREP3, derived from GenBank: AY028461.1 sequence information from M-line strain, Synthesized by GenScript), *Bg*TEP1 (r*Bg*TEP1, GenBank: HM003907.1, sequence information from Brazil strain, isolated from M-line strain) and *Bg*FREP2 (r*Bg*FREP2, GenBank: AY012700.1, sequence information from M-line strain, isolated from M-line strain) were generated by using the Gateway cloning system according to the manufacturer's instructions (Life Technologies) as previously described (*Hambrook et al., 2018*; *Pila et al., 2016a*; *Pila et al., 2017b*; *Pila et al., 2016b*), with primer information as shown in Table. S1 and Sf9 cell codon-optimized *Bg*FREP3 gene sequence information as shown in *Figure 1—figure supplement 1*. Briefly, the coding regions of *Bg*FREP3, *Bg*TEP1 and *Bg*FREP2 were amplified with Phusion high-fidelity DNA polymerase from targeted DNA templates in pUC57 plasmids (synthesized by GenScript) or cDNA generated by reverse transcription of mRNA derived from M-line strain. Blunt-end PCR products were cloned into the pENTR/D-TOPO vectors to generate entry clones. Plasmids from these entry

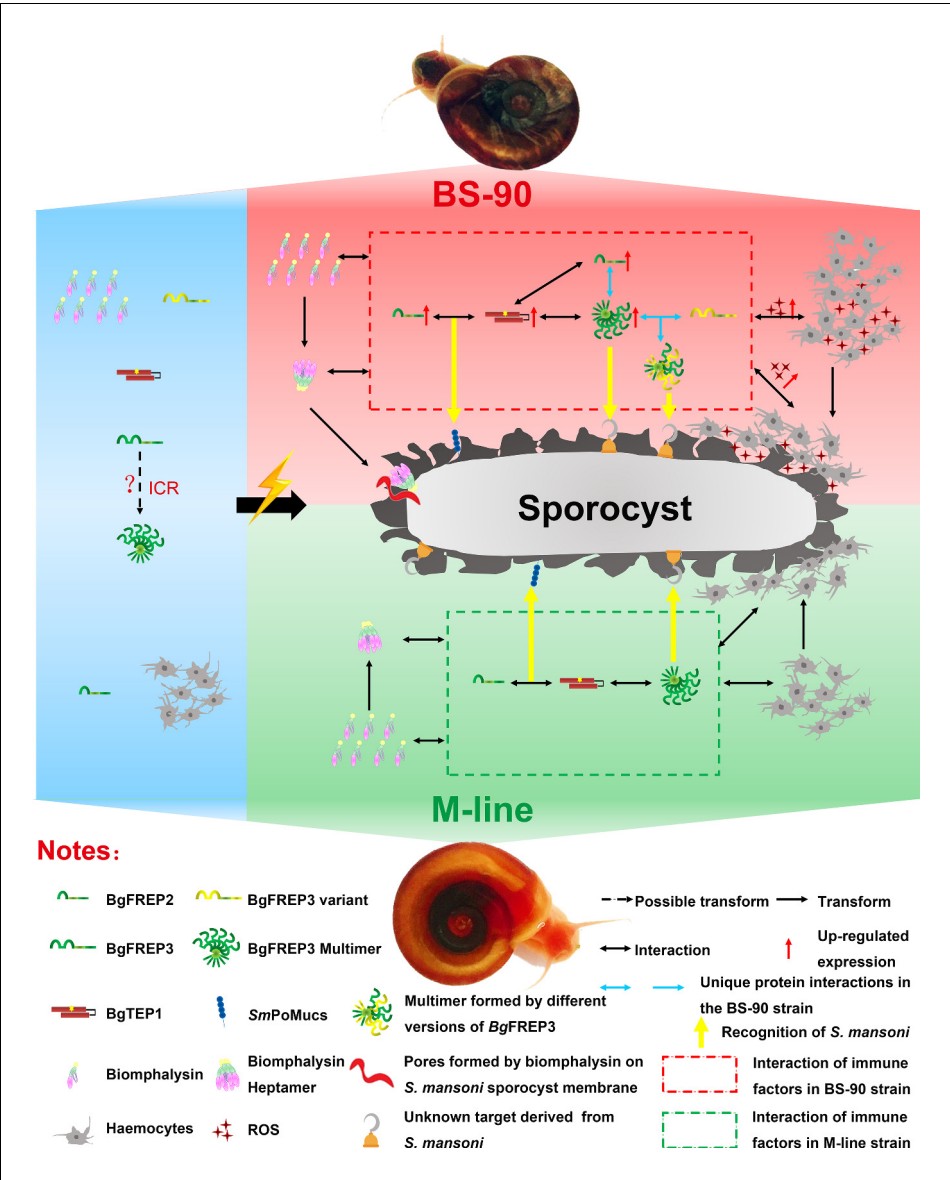

**Figure 5.** Interaction of *Bg*FREP3, *Bg*FREP2, *Bg*TEP1 and Biomphalysin in *B. glabrata* snail haemolymph and possible functional mechanisms against *S. mansoni* infection. This figure is a schematic drawing based on the results of previous studies and our work. In the absence of *S. mansoni* invasion (blue background area), both M-line and BS-90 *B. glabrata* snail hemolymphs contain *Bg*FREPs, *Bg*TEP1 and Biomphalysin and other immune factors. Upon *S. mansoni* infection, BS-90 strain (red background area) and M-line (green background area) strains demonstrate different protein interaction repertoires (red dotted frame in BS-90, green dotted frame in M-line). According to our pull-down results, *Bg*FREP3 in BS-90 plasma specifically interacts with *Bg*FREP2 and other versions of *Bg*FREP3 (blue horizontal one-way or two-way arrows). In addition, compared with the M-line strain, the BS-90 strain has more haemocytes (immune effector cells) in the snail haemolymph, and *Bg*FREP3, *Bg*FREP2, and *Bg*TEP1 are all up-regulated (red upward arrow). The more complex interactome of immune factors in BS-90 plasma results in higher levels of ROS (with cytotoxic activity) secreted by haemocytes. In conclusion, different coordination of humoral immune factors between BS-90 and M-line strains aids in dictating compatibility to *S. mansoni* infection.

clones were extracted and cloned into pIB/V5-His-DEST vectors in a Clonase recombination reaction to produce the expression clones. Recombinant plasmids were extracted from the expression clones and then transfected into Sf9 insect cells using Cellfectin reagent (Life Technologies). Cells that survived the screening of the antibiotic blasticidin (Thermo Fisher Scientific) were collected and lysed.

Protein expression of r*Bg*FREP3, r*Bg*TEP1 and r*Bg*FREP2 were detected by Western blot using monoclonal antibody against the V5 tag (Thermo Fisher Scientific) on the recombinant proteins.

Sf9 cells stably expressing recombinant r*Bg*FREP3, r*Bg*TEP1 and r*Bg*FREP2 were selected, some of them were frozen as a seed stock and the rest were maintained in a medium containing blasticidin. Cultures of Sf9 cells expressing recombinant proteins were scaled up, and the pellet (containing r*Bg*FREP3) or the medium (containing secreted r*Bg*TEP1 and r*Bg*FREP2) was collected. Recombinant proteins were purified from the cell pellet lysate or the culture medium with HisTrap FF (GE Healthcare) columns using fast protein liquid chromatography (ÄKTA Pure, GE Healthcare). Purified r*Bg*FREP3, r*Bg*TEP1 and r*Bg*FREP2 were dialyzed against PBS buffer twice for 2 hr each and then once overnight using Slide-A-Lyzer dialysis kit (Thermo Scientific).

## Preparation of *B. glabrata* plasma and pull-down experiment

Haemolymph was obtained from M-line and BS-90 strain *B. glabrata* snails by the head-foot retraction method (*Pila et al., 2016a*). Upon collection, haemolymph from ~5 snails of each strain (8–12 mm in diameter) was immediately placed in 1.5 mL tubes on ice, EDTA was added to a final concentration of 0.5 mM. Samples were centrifuged twice at 20,000 × g for 15 min and the cell-free fraction was collected. The 4 M Imidazole Stock Solution was added to a final concentration of 10 mM imidazole. Plasma samples were used immediately for subsequent pull-down experiment.

Pull-down assays were conducted by using Pull-Down PolyHis Protein:Protein Interaction Kit (Thermo Scientific) according to the manufacturer's instructions with some modifications. Briefly, first we prepared the 'bait' proteins, r*Bg*FREP3 and r*Bg*TEP1, from previously purified proteins or Sf9 cell lysate expressing r*Bg*FREP3 or concentrated medium containing r*Bg*TEP1. We then loaded the His-Pur Cobalt Resin equilibrated for at least 30 min at room temperature onto the column according to the kit instructions. The prepared polyhistidine-tagged r*Bg*FREP3 and r*Bg*TEP1 was immobilized on the columns after five times washing with wash solution [1:1 of Tris-buffered saline (TBS) solution: Pierce Lysis Buffer containing 10 mM imidazole]. For cell lysates or concentrated medium containing r*Bg*FREP3 and r*Bg*TEP1, protein expression was previously analyzed by SDS-PAGE and Western blot, and 800 µL was added; for purified r*Bg*FREP3, at least 150 µg (about 500 µL) was added. After incubation for 2 hr at room temperature or overnight at 4°C, the columns were washed 5–8 times with wash solution and then 800 µL of prepared plasma (prey) containing 10 mM imidazole was added to the columns. The columns were incubated at 4°C for overnight with gentle rocking motion every few hours. After that, the columns were washed 5–8 times with the wash solution, and then eluent (washing solution containing 290 mM imidazole) was added and incubated at room temperature for 30 min with gentle rocking on a rotating platform.

The eluent collected by centrifugation was heat-treated in a Laemmli loading buffer containing a reducing agent (β-mercaptoethanol) followed by SDS-PAGE analysis and silver staining (Pierce Silver Stain for Mass Spectrometry, Thermo Scientific) was performed according to the manufacturer's instructions. After silver staining, we excised the desired bands for LC-MS/MS to identify the proteins.

## Mass spectrometry analysis

Mass spectrometry and protein identification was completed by Alberta Proteomics and Mass Spectrometry Facility at University of Alberta. In-gel trypsin digestion was performed on the samples. Briefly, the excised gel bands were reduced (10 mM mercaptoethanol in 100 mM bicarbonate) and alkylated (55 mM iodoacetamide in 100 mM bicarbonate). After dehydration enough trypsin (6 ng/µL, Promega Sequencing grade) was added to just cover the gel pieces and the digestion was allowed to proceed overnight (~16 hr) at room temperature. Tryptic peptides were first extracted from the gel using 97% water/2% acetonitrile/1% formic acid followed by a second extraction using 50% of the first extraction buffer and 50% acetonitrile.

The digested samples containing tryptic peptides were resolved and ionized by using nanoflow HPLC (Easy-nLC II, Thermo Scientific) coupled to an LTQ Orbitrap XL hybrid mass spectrometer (Thermo Scientific). Nanoflow chromatography and electrospray ionization were accomplished by using a PicoFrit fused silica capillary column (ProteoPepII, C18) with 100 µm inner diameter (300 Å, 5 µm, New Objective). The mass spectrometer was operated in data-dependent acquisition mode, recording high-accuracy and high-resolution survey Orbitrap spectra using external mass calibration,

with a resolution of 30,000 and m/z range of 400–2000. The fourteen most intense multiply charged ions were sequentially fragmented by using collision induced dissociation, and spectra of their fragments were recorded in the linear ion trap; after two fragmentations all precursors selected for dissociation were dynamically excluded for 60 s. Data was processed using Proteome Discoverer 1.4 (Thermo Scientific) and a *B. glabrata* proteome database (UniProt) was searched using SEQUEST (Thermo Scientific). The proteome database was a genome assembly (Genome Accession: GCA_000457365) (*Adema et al., 2017*) , with download link https://www.uniprot.org/proteomes/UP000076420. Search parameters included a precursor mass tolerance of 10 ppm and a fragment mass tolerance of 0.8 Da. Peptides were searched with carbamidomethyl cysteine as a static modification and oxidized methionine and deamidated glutamine and asparagine as dynamic modifications.

## Sporocyst transformations of *S. mansoni* and immunofluorescence staining

Miracidia were obtained from eggs isolated from the livers of mice infected with *S. mansoni* (NMRI) as described previously (*Hambrook et al., 2018*). Briefly, newly hatched miracidia were collected for cultivation in Chernin's Balanced Salt Solution (CBSS) (*Chernin, 1963*) containing 1 g/L each of glucose and trehalose as well as 1% pen-strep antibiotics (Life Technologies). After cultivation for 24 hr at 26°C under normoxic conditions, most miracidia transformed to primary sporocysts. These experiments were divided into two different treatment methodologies, and carried out in parallel. One treatment method was the interaction of fixed sporocysts with recombinant proteins. In this method, sporocysts were washed five times with snail PBS (sPBS) (*Yoshino and Laursen, 1995*), grouped and transferred to 15 mL centrifuge tubes (Corning). All in-tube washes were performed by centrifugation for 2 min at 300 × g. Primary sporocysts were fixed with 2% paraformaldehyde/sPBS (pH 7.2) at 4°C overnight, followed by five washes with sPBS and 1 hr incubation in 5% BSA/0.02% azide/sPBS (blocking buffer) at room temperature. Sporocysts were then incubated with r*Bg*FREP3 (450 ng/μL), r*Bg*TEP1 (18 ng/μL) and r*Bg*FREP2 (70 ng/μL) in blocking buffer overnight at 4°C; while control groups were incubated in blocking buffer without recombinant proteins.

In the other treatment method, live primary sporocysts were collected and grouped, washed three times with CBSS, and directly incubated with recombinant proteins (concentrations the same as in the fixed treatment method) in a 26°C incubator. Sporocysts were then washed, fixed, washed, and blocked, as per the same steps as the previous treatment method.

From then on, sporocysts from both fixed and alive treatment methods were washed five times with 1% BSA/sPBS, sporocysts and then treated with anti-V5 antibody (Life Technologies) diluted 1:400 in blocking buffer. Following primary antibody treatment, sporocysts were washed five times with 1% BSA/sPBS and incubated with a solution containing 7.5 U/mL Alexa Fluor 488 phalloidin (Invitrogen) and 4 μg/mL Alexa Fluor 633 goat anti-mouse secondary antibody (Invitrogen) in blocking buffer overnight at 4°C. Finally, sporocysts were washed five times with 1% BSA/sPBS, given 25 μL of DAPI, incubated for 5 min at room temperature, and transferred into glass bottom microwell dishes (MatTeK, USA). Sporocysts were imaged using an Leica TCS SP5 laser scanning confocal microscope and analyzed using ImageJ (NIH) and Photoshop CS4 (Adobe Systems Inc, USA).

## Anti- *Bg*FREP3 polyclonal antibody generation and validation

The IgSF1 and FBG domains of *Bg*FREP3 (AAK28656.1) were screened for regions of presumed antigenicity using the Optimum Antigen design tool (GenScript). Three polyclonal antibodies were generated specific for different peptides (14–15 amino acids long,~4.4 kDa) targeting each domain. The information of these peptides is: anti-IgSF #1: CSFKKDDLSDSKQRS; anti-IgSF #2: CHNKYSEGRIDKSSN; anti-IgSF #3: CNINKDLDFKEQNIT; anti-FBG #1: TTFDRDNDEYSYNC; anti-FBG #2: GWKEYRDGFGDYNIC and anti-FBG #3: CLNGKWGSSDFAKGV. Peptides were synthesized by GenScript and used to immunize rabbits. Rabbits received a primary immunization and two secondary boosts at 2 and 5 weeks. One week after the second boost, rabbits were exsanguinated and the serum IgG was affinity-purified using protein A/G affinity column. These were further purified using affinity resin conjugated with the immunization peptides. The anti-*Bg*FREP3 antibodies used in this study were effective for Western blot detections at a concentration of 1:1000.

The purified antibodies were supplied by GenScript along with synthesized peptides and pre-immune serum. Specificity of the antibodies was tested against the respective peptides as well as snail plasma and haemocyte lysates. Dot blots were initially used to determine whether the antibodies recognized their peptides as well as the specificity of the recognition. Briefly, 2 µL of each supplied peptide was slowly pipetted in one spot on a nitrocellulose membrane that was divided into a grid with a pencil. The deposited peptides were dried at room temperature for about 10 min, creating a spot of approximately 2–3 mm. The membrane was then incubated in blocking buffer and processed using the same procedure described for Western blot following sample transfer to nitrocellulose membranes. In order to determine that the peptides were not being recognized by any pre-immune serum components, the above procedure was followed exactly with the exception that the respective pre-immune sera were used for the primary incubations in place of antibodies.

## Western and far-Western blot analysis

To test whether rBgTEP1 complexes with rBgFREP3 and rBgFREP2, purified rBgTEP1 was incubated with rBgFREP3 or rBgFREP2 in the following ratios (wt/wt in µg): 1:1, 1:0.75, 1:0.5, and 1:0.25 in 1.5 mL tubes for 2 hr at room temperature on a Labquake shaker (ThermoFisher Scientific) to allow potential complexes to form. After the incubation, samples were subjected to Western blot detection under denaturing (Laemmli protein loading buffer with β-mercaptoethanol and heated at 95°C for 10 min) and non-denaturing (no β-mercaptoethanol nor heating) conditions, respectively.

The Western blot experiment process was as described previously (*Pila et al., 2017b*). Samples were loaded on 6–14% (vol/vol) SDS-PAGE gels and run on the Mini PROTEAN Tetra system (Bio-Rad) at 150 V for 1.5–2 hr. Samples were then transferred for 1.5 hr onto 0.45 µm supported nitrocellulose membranes (Bio-Rad). Blocking was done for 1 hr at room temperature in 5% (wt/vol) skimmed milk or bovine serum albumin (BSA) prepared in TBS solution plus 0.1% Tween-20 (TBS-T buffer) before staining for 1 hr in anti-V5 mouse primary antibody at a dilution of 1:5000 in blocking buffer. Membranes were washed in TBS-T buffer for 10 min, then twice for 5 min each and once in TBS solution for 5 min. Membranes were then incubated for 1 hr in HRP-conjugated goat anti-mouse antibody diluted 1:5000 in blocking buffer followed by a wash step as described above. Finally, the blot was developed by incubating the membranes in SuperSignal West Dura Extended Duration substrate (Thermo Scientific). Chemiluminescent signals were acquired on the ImageQuant LAS 4000 machine (GE Healthcare).

To detect the BgFREP3 background expression in the *B. glabrata* snail plasma, the haemolymph of four snails was obtained from each of M-line and BS-90 strain *B. glabrata* snails. The haemolymph from each snail was diluted twice with sPBS buffer (containing 0.5 mM EDTA). Samples were centrifuged at 20,000 × g for 5 min and the supernatant (cell-free plasma) was transferred into new 1.5 mL tubes. After being centrifuged at 20,000 × g for 5 min again, the supernatant was collected and used for quantification. Samples (330 µg/well) were used for subsequent Western blot analysis. The primary antibodies were polyclonal antibodies raised in rabbits against the IgSF and FBG domains of rBgFREP3 (1:500 dilution) and the secondary antibody was a HRP-conjugated goat anti-rabbit IgG (1:5000 dilution).

Far-Western blotting is used to probe a membrane containing transferred protein with another protein to detect specific protein-protein interactions (*Wu et al., 2007*). In our study, we detected rBgTEP1 ('prey' protein) that were separated by SDS-PAGE and transferred onto a nitrocellulose membrane before being detected by biotinylated rBgFREP3 and rBgFREP2 ('bait' or 'probe' proteins) and a streptavidin-HRP chemiluminescent detection system. If rBgTEP1 forms a complex with rBgFREP3 or rBgFREP2, rBgFREP3 would be detected on spots in the membrane where rBgTEP1 was located. First, rBgFREP3 and rBgFREP2 was biotinylated with EZ-Link Sulfo-NHS-LC-Biotin (Thermo Scientific) according to the manufacturer's instructions. Gradually decreasing amounts (2.0, 1.3, 0.6 and 0 µg) of rBgFREP1 were separated by SDS-PAGE, and transferred to a membrane, as in a standard Western blot. Recombinant BgTEP1 in the membrane was then denatured and renatured as described previously (*Wu et al., 2007*). The membrane was then blocked and probed with biotinylated rBgFREP3 or rBgFREP2. After washing as described above in a standard Western blot, the membrane was incubated with streptavidin-HRP diluted 1:100,000 in blocking buffer at room temperature for 1 hr. Chemiluminescent signals were detected using the ImageQuant LAS 4000 machine (GE Healthcare).

## Quantitative real-time PCR analysis of *Bg*TEP1 expression

A TRIzol reagent was utilized to extract total RNA from five whole unchallenged snails from each strain independently, after which RNA was purified using a PureLink RNA mini kit (Life Technologies). A NanoVue spectrophotometer (ThermoFisher Scientific) was used to determine RNA concentration, and first-strand cDNA synthesis using the qScript cDNA synthesis kit (Quanta Biosciences) was performed using 1 μg of RNA template. The cDNA was diluted fivefold, and 5 μL was used as template in RT-PCR using primers (*Supplementary file 1*) specific for *Bg*TEP1 and *B. glabrata* β-actin (*Bg*Actin) in a SYBR Green detection system (PerfeCTa SYBR Green FastMix; Quanta Biosciences). All RT-PCRs were performed on a QuantStudio 3 PCR system (Applied Biosystems) using the following thermocycling conditions: initial hold at 95°C for 10 min, followed by 40 cycles of 95°C for 15 s and 60°C for 1 min, with data collection every cycle. Specificity for the gene specific primers RT-PCR was confirmed by continuous melt curve analysis.

## In vitro S. mansoni sporocyst killing assays

Haemolymph from 25 M-line or BS-90 strain *B. glabrata,* sterilized as reported in *Hahn et al. (2001)*, was pooled into 1.5 mL tubes and kept on ice. Haemocytes were isolated from plasma by centrifugation at 70 × g for 10 min at 4°C followed by aspiration of the cell free plasma, which was kept for assessment of plasma-mediated killing of sporocysts. Haemocytes were resuspended in modified *Bg*e-cell medium (m*Bg*e) medium [22% Schneider's *Drosophila* medium (Thermo Scientific), 7 mM D-glucose, 24 mM NaCl, and 20 mg/mL gentamicin at pH 7.4] and washed three times following the same centrifugation protocol. Haemocytes were then resuspended to a final concentration of 50 cells/μL in m*Bg*e medium and 200 μL of this suspension was added to the wells of a 96-well tissue culture plate pre-treated with poly-L-lysine that contained 10 *s. mansoni* sporocysts that were transformed using previously published protocols (*Hambrook et al., 2018*). Treatment consisted of 200 pM of each recombinant protein (*Bg*FREP3, *Bg*TEP1, *Bg*FREP2). Three anti-*Bg*FREP3 antibodies (anti-IgSF #1 and #3, anti-FBG #1) that were generated to peptides within the IgSF and FBG domains were used as a cocktail (combined concentration of 2.5 mg/mL) were used the antibody blocking assays. Sporocyst viability was assessed using the protocol published by *Hahn et al. (2001)*. Briefly, Propidium iodide (Millipore Sigma), was added to each well to a final concentration of 10 μg/mL. Sporocyst viability was assessed using a fluorescent inverted microscope (Zeiss) starting at 3 hr post incubation with haemocytes/plasma and then measured again at 6, 12, 24, 36 and 48 hr post incubation. Sporocyst death was determined by propidium iodide-staining of the nuclei of the sporocyst. Any sporocyst deemed dead, was left in the well until the end of the time-course so that the experimental conditions were not disturbed differentially in wells where sporocyst mortality was more frequent.

Cell-free plasma from M-line and BS-90 snails was ultracentrifuged at 30,000 rpm for 3 hr at 4°C to remove haemoglobin, which is known to have a toxic effect on sporocysts (*Bender et al., 2002*). Haemoglobin removal using ultracentrifugation has been shown to lead to no qualitative differences in plasma protein composition other than the depletion of haemoglobin (*Zelck et al., 1995*). Ultracentrifuged plasma was mixed 50:50 with m*Bg*e medium at room temperature and then incubated with 10 *s. mansoni* sporocysts according to the protocol outlined above.

## Assessment of reactive oxygen species production by haemocytes in vitro

To assess the role that ROS had on sporocyst killing by haemocytes the above haemocyte in vitro killing assays were repeated with addition of three well characterized inhibitors of ROS; diphenyleneiodonium chloride (DPI) and N-acetyl-L-cysteine (NAC) and catalase. All ROS inhibitors (DPI at a final concentration of 150 nM, NAC at a final concentration of 1 mM and a working concentration of 14000 units of catalase/mL) were incubated along with haemocytes (and plasma in the case of catalase) and *S. mansoni* sporocysts as described above in the haemocyte-based sporocyst killing assay.

## Bioinformatic and statistical analysis

We used Clustal Omega (https://www.ebi.ac.uk/Tools/msa/clustalo/) to align the amino acid sequences for *Bg*TEPs, Biomphalysins and *Bg*FREPs which acquired from GenBank [National Center for Biotechnology Information (NCBI)]. The alignment files were downloaded and visually inspected for

obvious inaccuracies, modified if need be, and then annotated based on information provided by NCBI. To show the distribution of the identified peptides over a full length sequence, the amino acid sequences of BgTEP1.5 (ADE45333.1), Biomphalysin (A0A182YTN9) and BgMFREP2 (AAK13550.1) were imported into Vector NTI Advance 11.5 (Thermo Fisher Scientific) and the identified peptides were annotated. Images produced by Vector NTI software were saved by FSCapture and further processed with FSCapture and Adobe Photoshop CS4. To determine significant differences in sporocysts killing assays, one-way ANOVA with Tukey's post hoc tests were performed using GraphPad Prism version 6.0 f for Mac OS X (GraphPad; www.graphpad.com). Statistical significance threshold was set at $p \leq 0.05$.

## Acknowledgements

B. glabrata snails provided by the NIAID Schistosomiasis Resource Center of the Biomedical Research Institute (Rockville, MD) through NIH-NIAID Contract HHSN272201700014I for distribution through BEI Resources. Microscopy was performed in the University of Alberta, Faculty of Medicine and Dentistry's Cell Imaging Core. These studies were supported by funds provided by the Natural Sciences and Engineering Research Council of Canada #2018–05209 and 2018–522661 (PCH), and National Natural Science Foundation of China (NSFC 31272682), Guangxi Natural Science Foundation for Young Scientists (2018JJB140423), Guangxi 16 Natural Science Foundation (Key Project, 2016JJD130059) Special Fund for Team Building, Beibu Gulf University (Former Qinzhou University, 2015).

## Additional information

### Funding

| Funder | Grant reference number | Author |
| --- | --- | --- |
| Natural Sciences and Engineering Research Council of Canada | 2018-05209 | Patrick Hanington |
| Natural Sciences and Engineering Research Council of Canada | 2018- 522661 | Patrick Hanington |
| National Natural Science Foundation of China | 31272682 | Xinzhong Wu |
| Natural Science Foundation of Guangxi Province | 2016JJD130059 | Xinzhong Wu |

The funders had no role in study design, data collection and interpretation, or the decision to submit the work for publication.

### Author contributions

Hongyu Li, Conceptualization, Formal analysis, Investigation, Methodology, Writing - original draft, Writing - review and editing; Jacob R Hambrook, Conceptualization, Formal analysis, Investigation, Methodology, Writing - review and editing; Emmanuel A Pila, Investigation, Methodology, Writing - review and editing; Abdullah A Gharamah, Methodology, Writing - review and editing; Jing Fang, Investigation, Methodology; Xinzhong Wu, Funding acquisition, Investigation, Methodology, Writing - review and editing; Patrick Hanington, Conceptualization, Supervision, Funding acquisition, Methodology, Project administration, Writing - review and editing

### Author ORCIDs

Patrick Hanington https://orcid.org/0000-0002-3964-5012

## Ethics

Animal experimentation: All animal work observed ethical requirements and was approved by the Canadian Council of Animal Care and Use Committee (Biosciences) for the University of Alberta (AUP00000057).

## Decision letter and Author response

Decision letter https://doi.org/10.7554/eLife.51708.sa1
Author response https://doi.org/10.7554/eLife.51708.sa2

## Additional files

### Supplementary files

• Supplementary file 1. Primer list for cloning and quantitative RT-PCR.

• Supplementary file 2. The identified peptides of *Bg*TEP1, Biomphalysin, *Bg*FREP2 and *Bg*FREP3.3 by LC-MS/MS.

• Transparent reporting form

### Data availability

All data generated or analysed during this study are included in the manuscript and supporting files.

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
