## [Decision Letter]

**Acceptance summary:**

Schistosomiasis, a disease caused by parasitic trematodes of the genus *Schistosoma*, is the second-most socioeconomically devastating parasitic disease. *Schistosoma* use snails as intermediate hosts for transmission to a vertebrate host. The molecular interactions between the snail (*Biomphalaria glabrata*) and *Schistosoma (Schistosoma mansoni*) have been studied extensively towards better understanding this essential life cycle stage of an important human parasite. Some B. glabrata snails are naturally resistant to *Schistosoma* infection while other are susceptible. Researchers have long sought immune determinants present in resistant snail that specifically activate haemocytes to encapsulate and destroy *Schistosoma*. This led to the identification of soluble lectins termed fibrinogen-related proteins (*Bg*FREPs). This article describes the underlying mechanism of *Bg*FREP function reporting how it binds to Schistosoma *mansoni* sporocyst surfaces, and then how recognition is translated into haemocyte engagement, activation, and ultimately parasite encapsulation. Collectively, it provides insight into how numerous previously characterized immune factors known to be important in the anti-*Schistosoma* immune response act in concert to defend the snail host. This an important contribution in the fields on invertebrate immunity and parasitology.

**Decision letter after peer review:**

Thank you for submitting your article "Coordination of humoral immune factors dictates compatibility between < *Schistosoma mansoni* and *Biomphalaria glabrata*" for consideration by *eLife*. Your article has been reviewed by three peer reviewers, and the evaluation has been overseen by a Reviewing Editor and Satyajit Rath as the Senior Editor. The following individuals involved in review of your submission have agreed to reveal their identity: Coen Adema (Reviewer #2); Tim Yoshino (Reviewer #3).

The reviewers have discussed the reviews with one another and the Reviewing Editor has drafted this decision to help you prepare a revised submission.

Summary:

Although a diversity of soluble hemolymph proteins has been previously associated with immune resistance to larval schistosome infection in inbred strains of the snail *Biomphalaria glabrata*, the mechanism(s) mediating larval recognition and/or killing in model *B. glabrata/S. mansoni* systems has not been defined. Li et al. present evidence that differing molecular interaction of four of the major immune proteins (Frep3, Tep1, Frep2 and Biomphalysin) can account for or explain the infectivity phenotypes exhibited by susceptible (M-line) and resistant (BS-90) *B. glabrata* strains. This was accomplished by generating recombinant Frep3, Tep1 and Frep2 proteins and specific anti-Frep3 antisera, and incorporating these in plasma pull-down assays, sporocyst binding assays, Western and far-Western blot analyses and in vitro cytotoxicity inhibition assays. All experiments appeared to be well-designed and appropriately controlled, and overall results generally support their conclusion that interactions between Frep3, in association with Tep1 and Frep2, and host hemocytes account for the different compatibility phenotypes exhibited in this model system. However, there are several points that need addressing that could result in altered conclusions or significant clarifications.

Essential revisions:

1) Physical evidence that rFrep3 alone, but not rFrep2 or rTep1 alone, can bind to sporocysts is provided by ICC using fixed parasites. Assuming, as stated, that recomb. proteins bind to the outer tegumental of larvae, the localization pattern of Frep3 and Frep2/Tep1 binding do not appear to reflect such a surface binding pattern. This raises the question regarding the specificity of Frep2/3/Tep reactivity with the tegument of living sporocysts, which is at the center of this paper. Performing similar ICC experiments, initially treating live sporocysts to recombinant proteins (suggest shorter exposure times of 4-6 hrs), washing and then fixing, should provide a more accurate representation of Frep/Tep binding interactions with the unaltered sporocyst surface.

2) Biomphalysin, a pore-forming protein, is speculated as the mechanism responsible for plasma-mediated killing of sporocysts in vitro (Figure 4A). However, an alternative mechanism may be the presence of ROS (e.g., H_2_O_2_) naturally occurring in the plasma of BS-90 compared to M-line snails. To resolve this question, an experiment in which a ROS scavenger enzyme (e.g., catalase) is added to BS-90 plasma (and M-line plasma treated Frep/Tep) prior to sporocyst exposure in the CMC assay could be performed to eliminate an ROS-mediated mechanism.

3) "These pull-down results" Also with interaction taking place without involvement of parasite products, is this the same for BS90 and M snails? How is this regulated in absence of pathogen to avoid aspecific continuous immune activation? Does FREP3 transport TEP to recognized parasite antigens, resulting in activation of TEP to facilitate downstream activation?

---

## [Author Response]

Essential revisions:1) Physical evidence that rFrep3 alone, but not rFrep2 or rTep1 alone, can bind to sporocysts is provided by ICC using fixed parasites. Assuming, as stated, that recomb. proteins bind to the outer tegumental of larvae, the localization pattern of Frep3 and Frep2/Tep1 binding do not appear to reflect such a surface binding pattern. This raises the question regarding the specificity of Frep2/3/Tep reactivity with the tegument of living sporocysts, which is at the center of this paper. Performing similar ICC experiments, initially treating live sporocysts to recombinant proteins (suggest shorter exposure times of 4-6 hrs), washing and then fixing, should provide a more accurate representation of Frep/Tep binding interactions with the unaltered sporocyst surface.

Thank you for the advice. Initially, we did not use confocal microscopy to evaluate tegumental binding. In our revised submission, we confirm tegument binding by clearly demonstrating a ring of recognition around the sporocyst. We approached this study as suggested, incubating live sporocysts with r*Bg*FREP3, r*Bg*FREP2, r*Bg*FREP2+r*Bg*TEP1 and r*Bg*TEP1 along with controls. As is now indicated in the Materials and methods, live primary sporocysts of *S. mansoni* were incubated with recombinant proteins at 26°C, then washed and fixed, and the subsequent steps were consistent with previous immunofluorescence staining experiments. We also replicated the method of our initial study, adding ther*Bg*FREP3+r*Bg*TEP1 treatment to further validate our pull-down results that suggest that FRE3 and TEP associate without immune stimulation, and assessed using a confocal microscope. These repeated studies also yielded clear surface binding. To replace the original images, we chose to use fixed sporocysts because the fluorescence intensity with fixed sporocysts was stronger than that with live sporocysts. We have included the images of the sporocysts that were incubated with the treatments while alive (as was suggested) as Figure 2—figure supplement 1.

2) Biomphalysin, a pore-forming protein, is speculated as the mechanism responsible for plasma-mediated killing of sporocysts in vitro (Figure 4A). However, an alternative mechanism may be the presence of ROS (e.g., H_2_O_2_) naturally occurring in the plasma of BS-90 compared to M-line snails. To resolve this question, an experiment in which a ROS scavenger enzyme (e.g., catalase) is added to BS-90 plasma (and M-line plasma treated Frep/Tep) prior to sporocyst exposure in the CMC assay could be performed to eliminate an ROS-mediated mechanism.

We hadn’t considered the possibility that BS-90 plasma may contain naturally-occurring ROS that was underpinning plasma-mediated killing. We undertook the recommended study to utilize a ROS scavenger specific to H_2_O_2_, added in advance to the BS-90 plasma. A ROS scavenger (catalase) was pre-incubated with cell-free BS-90 plasma and then the rest of the study was performed following the previously established method. With the exclusion of H_2_O_2_, BS-90 plasma was still able to kill *S. mansoni* sporocysts, thereby strengthening the likelihood of a non-ROS mediated mechanism by which humoral killing is mediated. The new results are presented in a revised Figure 4—figure supplement 1B.

3) "These pull-down results" Also with interaction taking place without involvement of parasite products, is this the same for BS90 and M snails? How is this regulated in absence of pathogen to avoid aspecific continuous immune activation? Does FREP3 transport TEP to recognized parasite antigens, resulting in activation of TEP to facilitate downstream activation?

Regarding the r*Bg*FREP3 pull-down results, we identified *Bg*TEP1 in both BS-90 and M-line snails without the involvement of parasite products. Unfortunately, we do not know whether these interactions differ regarding their frequency within the different strains. Given that we have found *Bg*TEP1 levels to be similar between strains (Figure 1—figure supplement 3), the noticeable difference in *Bg*FREP3 levels, as evidenced by our Western Blots of snail plasma (Figure 1B), may in fact result in higher circulating levels of a *Bg*TEP/*Bg*FREP3 complex.

The reviewers raise an excellent point in questioning the mechanism of *Bg*TEP1 activation. Although *Bg*FREP3 appears to interact with *Bg*TEP1 in the plasma of both snail strains, as is demonstrated by the r*Bg*FREP3 pull-down results, we also show that r*Bg*FREP3 specifically interacted with *Bg*FREP2 and other versions of r*Bg*FREP3 only in BS-90 snail plasma. These differing results between strains suggests that specific immune complexes that associate with *Bg*TEP1 may be differentially activated and/or regulated depending on the specific *Bg*FREPs present. To date, we are not sure of the exact mechanism by which *Bg*TEP1 is activated, nor do we know how this activation is regulated in the absence of pathogen. In pursuit of a resolution to this query, we hypothesized that *Bg*TEP1 may operate in a similar manner to TEP1 of the *Anopheles gambiae*, in which a pair of Leucine-rich repeat containing molecules stabilize *Ag*TEP, prior to its binding to the surface of *Plasmodium* ookinetes (Fraiture et al., 2009). We thought that *Bg*FREP3 may function analogously to the LRR molecules LRIM1 and APLC, serving to stabilize *Bg*TEP1, while also guiding it to specific targets recognized by the *Bg*FREP on the parasite surface. Further protein pull-down investigations with a focus on determining whether *Bg*TEP1 cleavage products were observed reflected our past results, suggesting that *Bg*FREP3 interacts with a full-length form of *Bg*TEP1. This contrasts the observations from the *Anopheles* model and also those of Mone et al. which suggest that the C-terminal portion of *Bg*TEP1 associates with *S. mansoni* polymorphic mucins (*Sm*PoMucs). This suggests that once brought in association with the parasite, *Bg*TEP1 becomes cleaved/activated, but that activation (via cleavage into the N and C-terminal domains) does not occur without some pathogen-related stimulation. We have added clarification of this in the text of our Discussion.